# Mitochondrial fusion is required for spermatogonial differentiation and meiosis

Grigor Varuzhanyan[1], Rebecca Rojansky[1], Michael J Sweredoski[2], Robert LJ Graham[2†], Sonja Hess[2‡], Mark S Ladinsky[1], David C Chan[1]*

[1]Division of Biology and Biological Engineering, California Institute of Technology, Pasadena, United States; [2]Proteome Exploration Laboratory of the Beckman Institute, California Institute of Technology, Pasadena, United States

**Abstract** Differentiating cells tailor their metabolism to fulfill their specialized functions. We examined whether mitochondrial fusion is important for metabolic tailoring during spermatogenesis. Acutely after depletion of mitofusins *Mfn1* and *Mfn2*, spermatogenesis arrests due to failure to accomplish a metabolic shift during meiosis. This metabolic shift includes increased mitochondrial content, mitochondrial elongation, and upregulation of oxidative phosphorylation (OXPHOS). With long-term mitofusin loss, all differentiating germ cell types are depleted, but proliferation of stem-like undifferentiated spermatogonia remains unaffected. Thus, compared with undifferentiated spermatogonia, differentiating spermatogonia and meiotic spermatocytes have cell physiologies that require high levels of mitochondrial fusion. Proteomics in fibroblasts reveals that mitofusin-null cells downregulate respiratory chain complexes and mitochondrial ribosomal subunits. Similarly, mitofusin depletion in immortalized spermatocytes or germ cells in vivo results in reduced OXPHOS subunits and activity. We reveal that by promoting OXPHOS, mitofusins enable spermatogonial differentiation and a metabolic shift during meiosis.

*For correspondence:
dchan@caltech.edu

Present address: †School of Biological Sciences, Queens University Belfast, Medical Biology Centre, Belfast, United Kingdom; ‡Antibody Discovery and Protein Engineering, MedImmune, Gaithersburg, United States

Competing interests: The authors declare that no competing interests exist.

## Introduction

The first mitochondrial fusion gene, *fuzzy onions (fzo)*, was discovered in 1997 in a *Drosophila melanogaster* mutant exhibiting male sterility (*Hales and Fuller, 1997*). During early spermatid development in flies, mitochondria aggregate and fuse to become two giant organelles that wrap around each other to form the Nebenkern (*Fuller, 1993*). Mutations in *fzo* abolish mitochondrial fusion in spermatids, resulting in a morphologically aberrant Nebenkern and sterility. The reason for this stage-specific fusion event and how mitochondrial fusion promotes male fertility in *Drosophila* are unknown.

*Fzo* is the founding member of an evolutionarily conserved family of mitochondrial fusion proteins known as the mitofusins (Mfns). In humans, there are two homologs known as *MFN1* and *MFN2* (*Santel and Fuller, 2001*). Fusion acts as a major quality control mechanism for mitochondria by enabling mixing of matrix components and promoting their homogenization (*Chan, 2012*). In the absence of fusion, mitochondria diverge, become functionally heterogenous, and mitochondrial DNA (mtDNA) and oxidative phosphorylation (OXPHOS) are diminished (*Chen et al., 2005*; *Chen et al., 2003*; *Chen et al., 2007*; *Chen et al., 2010*; *Weaver et al., 2014*). Mitochondrial fusion also affects mitochondrial transport and degradation (*Chen et al., 2003*; *Gomes et al., 2011*; *Kandul et al., 2016*; *Misko et al., 2010*; *Rambold et al., 2011*).

In mammals, spermatogenesis is a cyclical process that involves differentiation of spermatogonia into spermatocytes, which undergo meiosis to form haploid spermatids and ultimately spermatozoa (*Griswold, 2016*). Throughout this process, germ cells differentiate in intimate association with

nursing Sertoli cells. In addition to providing differentiation cues and metabolites for the developing germ cells, Sertoli cells form the blood-testis barrier (BTB) that separates the seminiferous epithelium into the basal (towards the periphery) and apical (towards the lumen) compartments (*Stanton, 2016*). Spermatogonia reside within the basal compartment and are comprised of both undifferentiated and differentiating cells. Undifferentiated spermatogonia constitute a dynamic and heterogeneous population that includes the self-renewing stem cell pool (*de Rooij, 2017*; *Lord and Oatley, 2017*). Differentiating spermatogonia give rise to spermatocytes that cross the BTB and complete meiosis. After two meiotic divisions, each spermatocyte produces four haploid spermatids that transform into the specialized sperm cells capable of fertilization.

Several observations in humans and mice illustrate the importance of mitochondrial function during spermatogenesis. Some patients with mtDNA disease have sperm defects (*Demain et al., 2017*; *Folgerø et al., 1993*), and sperm from some infertile males harbor mtDNA mutations (*Baklouti-Gargouri et al., 2014*; *Carra et al., 2004*; *Kao et al., 1995*; *Lestienne et al., 1997*). Mouse models with a pathogenic mtDNA deletion exhibit spermatogenic arrest during the zygotene stage of Meiotic Prophase I (MPI) (*Nakada et al., 2006*). Furthermore, a mouse model that is unable to utilize mitochondrial ATP exhibits spermatogenic arrest during the leptotene stage of MPI (*Brower et al., 2009*). Finally, mouse models that accumulate mtDNA mutations exhibit male infertility (*Jiang et al., 2017*; *Kujoth et al., 2005*; *Trifunovic et al., 2004*). Much less is known about the role of mitochondrial dynamics in male fertility. The *Drosophila* homolog of mitofusin (*dMfn*) was recently shown to regulate lipid homeostasis and maintenance of germline stem cells in the testis (*Sênos Demarco et al., 2019*), A clinical study found that low Mfn2 expression in sperm is associated with asthenozoospermia (reduced sperm motility) and reduced sperm mitochondrial membrane potential (*Fang et al., 2018*). Finally, Mfn1 is required for spermatogenesis in the mouse (*Zhang et al., 2016*), but the precise stage of the defect as well as the role of Mfn2 remain unknown. To clarify the role of mitochondrial fusion during male germ cell development, we deleted *Mfn1, Mfn2,* and both *Mfn1* and *Mfn2* from the male germline and examined all stages of spermatogenesis. Our results show that mitochondrial fusion is required for spermatogonial differentiation and a metabolic shift during meiosis.

## Results

### Mitofusins are essential for mouse spermatogenesis

To investigate the role of mitofusins during male germ cell development, we removed *Mfn1, Mfn2,* or both *Mfn1* and *Mfn2* from the male germline by combining the previously described conditional alleles of *Mfn1* and *Mfn2* with the male germline-specific *Stra8-Cre* driver (*Chen et al., 2003*; *Chen et al., 2007*; *Sadate-Ngatchou et al., 2008*). We designate these mice as S8::Mfn1, S8::Mfn2, and S8::Dm (*double mutant*), respectively, and compared them with age-matched, wild-type (WT) littermates. Our mating scheme also incorporated the conditional *Rosa26^{PhAM}* allele, which encodes a mitochondrially-targeted, photo-activatable fluorescent protein, mito-Dendra2 (*Pham et al., 2012*). mito-Dendra2 served as a *Cre* reporter to label the mitochondrial matrix selectively in germ cells. With histological analysis of testis sections, we verified that mito-Dendra2 is restricted to the male germline and absent from the intimately associated Sertoli and interstitial cells (*Figure 1—figure supplement 1*). *Stra8-Cre* expression is reported to begin at post-natal day 3 (P3) in undifferentiated spermatogonia (*Sadate-Ngatchou et al., 2008*), including the majority of early stem-like GFRα1-positive spermatogonia (*Hobbs et al., 2015*). Consistent with this, our examination of the mito-Dendra2 Cre reporter clearly demonstrated excision in all germ cell types, including the vast majority of GFRα1-expressing spermatogonia (*Figure 1—figure supplement 2*).

All three mitofusin-deficient mouse lines were healthy and showed no changes in weight (*Figure 1—figure supplement 3*). However, they had obviously smaller testes compared with controls (*Figure 1A and B*), suggesting an essential role for mitochondrial fusion during spermatogenesis. Indeed, there is significant reduction of spermatozoa in the epididymides of S8::Mfn1 and S8::Mfn2 mice, with the defect more severe with loss of *Mfn1* (*Figure 1C and D*). The residual spermatozoa in both mutant lines often display mitochondrial fragmentation and reduced mitochondrial content (*Figure 1E and F*). Mutant spermatozoa also exhibit morphological defects, particularly kinking near or in the midpiece (*Figure 1E and G*), and almost a complete loss of motility (*Figure 1H*; *Videos 1*–

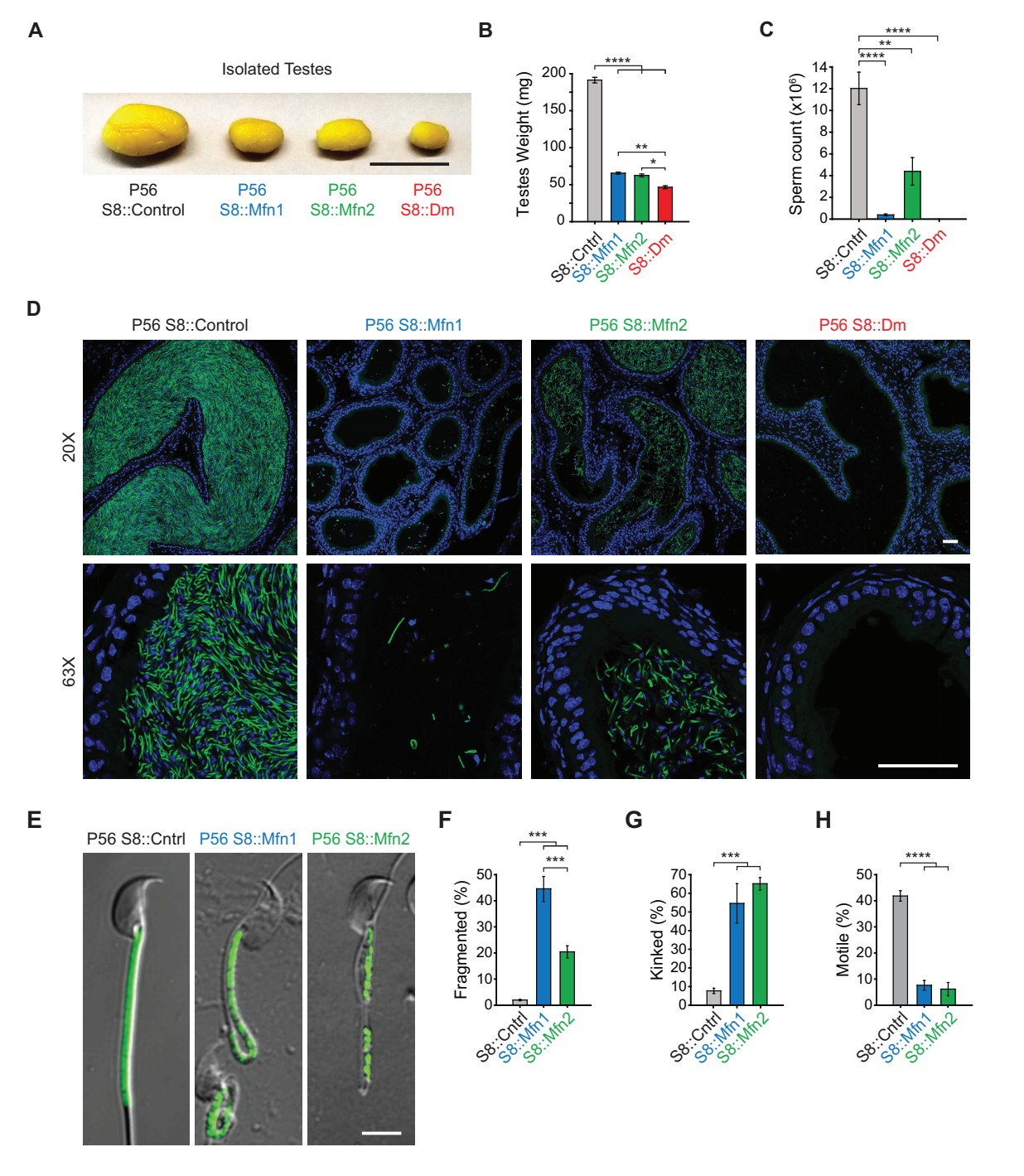

**Figure 1.** Mitofusins are essential for mammalian spermatogenesis. (A) Comparison of testicular size in mice of the indicated genotype. Tissues have been fixed with Bouin's solution, which gives them the yellow appearance. Scale bar, 0.5 cm. (B) Testes weight measurements from adult P56 mice. Both organs were used. N ≥ 7. (C) Epididymal sperm count. Note that S8::Dm mice do not produce any sperm. N ≥ 6. (D) Histological analysis of sperm in cauda epididymis sections. Mature sperm are identified by the rod-like mito-Dendra2 (Dn) signal (green) in the midpiece. Nuclei are labeled with DAPI (blue). Scale bars, 50 μm. (E) Morphological comparison of isolated caudal spermatozoa. Mitochondria are labeled with mito-Dendra2. Note that mutant sperm have patchy, fragmented mitochondria and sharp kinking near or in the midpiece. Scale bar, 5 μm. (F–H) Quantification of mitochondrial morphology (F), sperm morphology (G), and motility (H). N ≥ 3. For more information on motility, see *Videos 1–3*. All data are from adult P56 mice. Data are represented as mean ± SEM. ****p≤0.0001; ***p≤0.001; **p≤0.01; *p≤0.05. For statistical tests used, see Material and Methods section.

The online version of this article includes the following figure supplement(s) for figure 1:

*Figure 1 continued on next page*

*Figure 1 continued*

**Figure supplement 1.** mito-Dendra2 (Dn) expression in testis sections.
**Figure supplement 2.** Stra8-Cre is expressed in GFRα1-expressing spermatogonia.
**Figure supplement 3.** Quantification of animal weight from the indicated mouse lines.

*3*). S8::Dm mice have the smallest testes (*Figure 1A and B*) and strikingly, a complete absence of epididymal spermatozoa (*Figure 1C and D*). These results indicate an essential role for mitofusins in mammalian spermatogenesis.

## Mitofusins are essential for meiosis

To identify the specific stage(s) of the spermatogenic defect in mitofusin-deficient mice, we first analyzed juvenile mice at P24, before the completion of the first round of spermatogenesis. Even at this early time point, mutant mice have substantially smaller testes compared with control (*Figure 2A*). We performed Periodic Acid-Schiff (PAS) staining in testis sections and found that mutant seminiferous tubules are narrower in diameter, sparsely populated, and contain numerous Sertoli cell vacuolizations (*Figure 2B*). Whereas most control seminiferous tubules are packed with round spermatids, S8::Dm tubules have almost a complete absence of post-meiotic spermatids, indicative of an inability to complete meiosis.

To confirm this meiotic defect, we categorized all germ cell types using molecular markers in PFA-fixed, frozen sections (*Figure 2C–2E* and *Figure 2—figure supplement 1*). Haploid spermatids were identified with an antibody against SP-10 (*Osuru et al., 2014*), which marks the spermatid acrosome (*Figure 2C*). Spermatocytes in Meiotic Prophase I (MPI) were identified using an antibody against histone γH2AX (*Hamer et al., 2003*), which distinctively labels the various substages of MPI. Differentiated and undifferentiated spermatogonia were labeled using antibodies against c-Kit (*Rossi, 2013*) and PLZF (*Buaas et al., 2004*; *Costoya et al., 2004*), respectively. First, we scored seminiferous tubules based on the most differentiated cell type they contained (*Figure 2D* and S2A). As expected for the P24 time point, the majority of control seminiferous tubules (87 ± 6%) contained germ cells differentiated up to the spermatid stage, and the remaining tubules (13 ± 6%) contained cells differentiated up to pachytene/diplotene. In contrast, the majority of seminiferous tubules from P24 S8::Dm mice contained cells differentiated only up to the leptotene/zygotene stage (55 ± 2%), indicating a meiotic defect during early MPI. While some mutant seminiferous tubules contained cells differentiated up to the pachytene (22 ± 3%) and spermatid (14 ± 14%) stages (*Figure 2D*), these cell types were

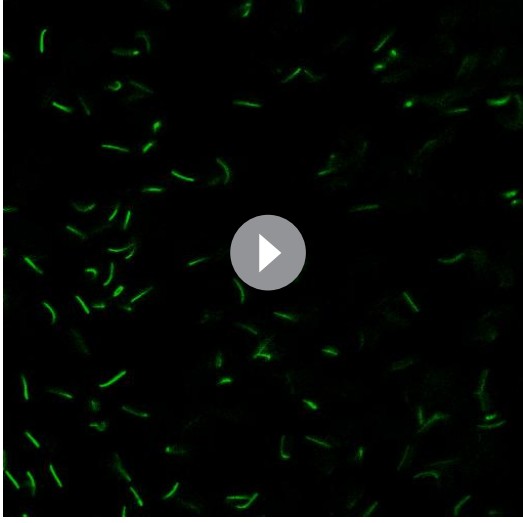

**Video 1.** Movie of sperm isolated from P56 S8::Control mice.
https://elifesciences.org/articles/51601#video1

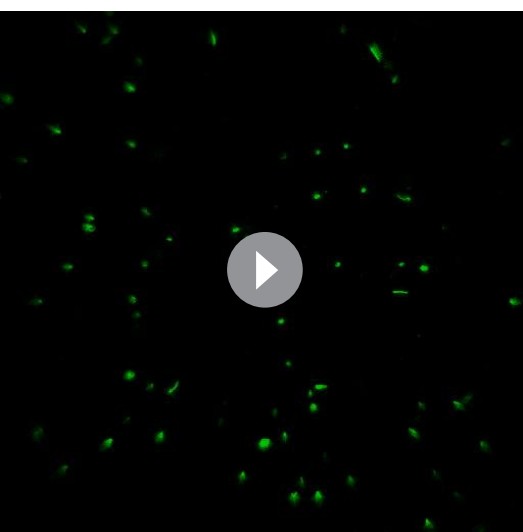

**Video 2.** Movie of sperm isolated from P56 S8::Mfn1 mice.
https://elifesciences.org/articles/51601#video2

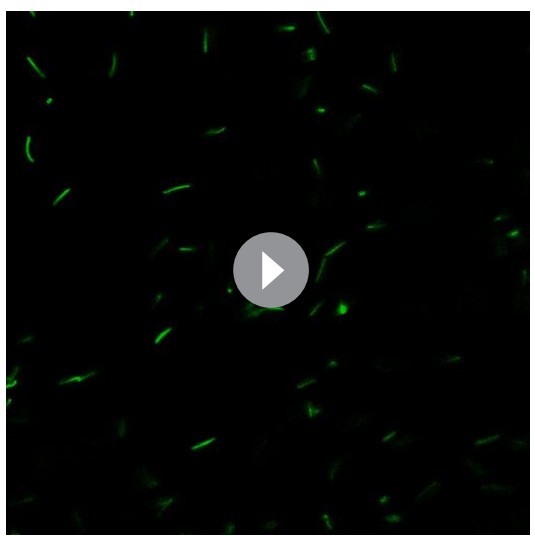

**Video 3.** Movie of sperm isolated from P56 S8::Mfn2 mice.
https://elifesciences.org/articles/51601#video3

depleted to less than 20% compared with control (*Figure 2E*).

To better characterize the meiotic defect, we performed chromosomal spreading of isolated germ cells and scored with γH2AX and SCP3, which mark double stranded breaks (DSBs) and the synaptonemal complex that forms between homologous chromosomes, respectively. Using this assay, we could visualize individual chromosomes and thus distinguish between all substages of MPI: leptotene, zygotene, pachytene, and diplotene (*Figure 2F*). Consistent with data from testis sections, we found that S8::Dm mice have a reduction of pachytene and diplotene spermatocytes and an abundance of both leptotene and zygotene cells, pinpointing the meiotic defect to the zygotene-to-pachytene transition (*Figure 2G*). There were no obvious chromosomal abnormalities in S8::Dm mice. To check whether loss of either mitofusin alone was sufficient to block meiosis, we performed histology and chromosomal spreading in S8::Mfn1 and S8::Mfn2 mice (*Figure 2—figure supplement 2*). We found that loss of *Mfn1* alone was sufficient to cause a similar defect in meiosis, resulting in accumulation of spermatocytes in the leptotene and zygotene stages, and reduced production of spermatids. In contrast, S8::Mfn2 mice had normal progression through MPI at P24. Because *Stra8-Cre*-mediated excision occurs in all germ cell types, including spermatogonia that give rise to meiotic spermatocytes, our results demonstrate that MPI is the most sensitive stage of spermatogenesis to disruption of mitochondrial fusion (*Figure 2H*).

## Mitofusins are required for a metabolic shift during meiosis

Given the MPI abnormality in fusion-deficient mice, we wondered whether mitochondria show any special features during this stage of spermatogenesis. We quantified mitochondrial content in meiotic spermatocytes by measuring mito-Dendra2 fluorescence in immunolabeled testis sections. In P24 WT mice, we found that pachytene/diplotene cells contained over three times more mitochondrial signal compared with leptotene/zygotene spermatocytes (*Figure 3A and B*). Pachytene/diplotene spermatocytes are much sparser in S8::Dm mutant animals, but when present, these cells have less than half the mitochondrial content found in WT animals (*Figure 3B*). To better visualize mitochondrial morphology during MPI, we performed meiotic spreading and immunostained against the outer mitochondrial membrane marker Tom20. We found that WT pachytene/diplotene spermatocytes, compared with leptotene/zygotene spermatocytes, contain elongated and highly clustered mitochondria (*Figure 3C and D*). Thus, spermatocytes in P24 S8::Dm mice arrest in spermatogenesis just before the increased mitochondrial biogenesis, elongation, and clustering that occurs in pachytene/diplotene. Furthermore, S8::Dm spermatocytes that do make it to the pachytene/diplotene stage have less mitochondrial content and their mitochondria are more fragmented (*Figure 3C and D*).

As spermatocytes initiate meiosis, they cross the blood-testis-barrier (BTB) and become dependent on lactate secreted by Sertoli cells for energy production (*Boussouar and Benahmed, 2004*). Lactate is oxidized to pyruvate and transported into mitochondria to fuel OXPHOS (*Vanderperre et al., 2016*). We therefore examined whether meiotic spermatocytes, relative to spermatogonia, have higher expression of the mitochondrial pyruvate carrier, MPC1. Indeed, we find that while pachytene/diplotene spermatocytes have strong expression of MPC1, spermatogonia have little to no MPC1 protein (*Figure 3E*).

The increased mitochondrial mass and upregulation of MPC1 in pachytene/diplotene spermatocytes suggest that spermatocytes have high OXPHOS utilization compared with pre-meiotic spermatogonia. To test this directly, we used established flow cytometry methods (*Bastos et al., 2005*) to isolate diploid spermatogonia and tetraploid MPI spermatocytes from WT dissociated

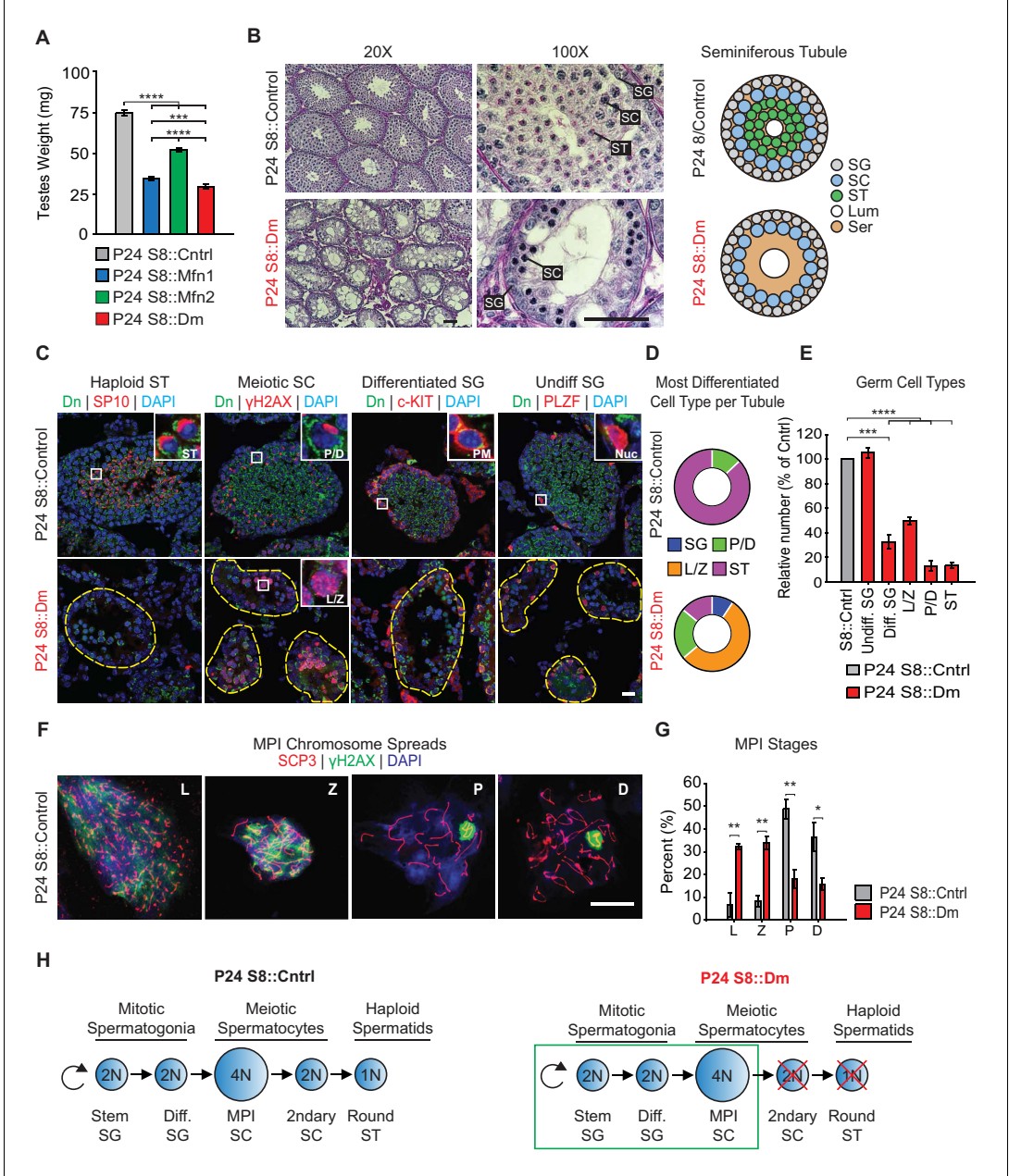

**Figure 2.** Mitofusins are required for meiosis. (**A**) Testes weight measurements from juvenile P24 mice. N ≥ 3. (**B**) Periodic Acid-Schiff (PAS) histology testis sections. Note the absence of post-meiotic spermatids in S8::Dm tubules. Scale bars, 50 μm. On the right are stylized depictions of the 100X panels. SG, spermatogonium; SC, spermatocyte; ST, spermatid, Lum, lumen; Ser, Sertoli cell cytoplasm. (**C**) Analysis of the major germ cell types in WT and mutant testis sections. For clarity, the borders of mutant seminiferous tubules are outlined by dashed lines. The markers used are indicated. Scale bar, 20 μm. Dn, mito-Dendra2. (**D**) Doughnut graphs tabulating the most differentiated cell type found in seminiferous tubule cross sections. For *p*-values, see *Figure 2—figure supplement 1*, which shows the same data displayed as bar charts. N = 4. (**E**) Quantification of germ cells in seminiferous tubule cross sections. Mutant values are plotted relative to control, which is set at 100% and indicated by the gray bar. There is an upward trend from differentiated spermatogonia to the leptotene/zygotene stage, but this difference was not statistically significant. N = 4. (**F**) Representative images of chromosomal spreads from WT spermatocytes in meiotic prophase I (MPI). Mutant meiotic spreads had no obvious chromosomal abnormalities. Scale bar, 20 μm. (**G**) Quantification of MPI substages from chromosomal spreads. In the mutant, note the bottleneck at the zygotene-to-pachytene transition. N = 4. (**H**) Schematic of normal spermatogenesis (left), and the meiotic defect in mutants (right). ST, spermatid; MPI, Meiotic Prophase I; SC, spermatocyte; P/D, pachytene/diplotene; L/Z, leptotene/zygotene; SG, spermatogonium. All data are from P24 juvenile mice. Data are represented as mean ± SEM. ****p≤0.0001; ***p≤0.001; **p≤0.01; *p≤0.05. For statistical tests used, see the Materials and methods section.

The online version of this article includes the following figure supplement(s) for figure 2:

**Figure supplement 1.** Same data as in *Figure 2D* but displayed as bar charts to indicate.

*Figure 2 continued on next page*

**Figure supplement 2.** Mfn1, but not Mfn2, is required for meiosis.

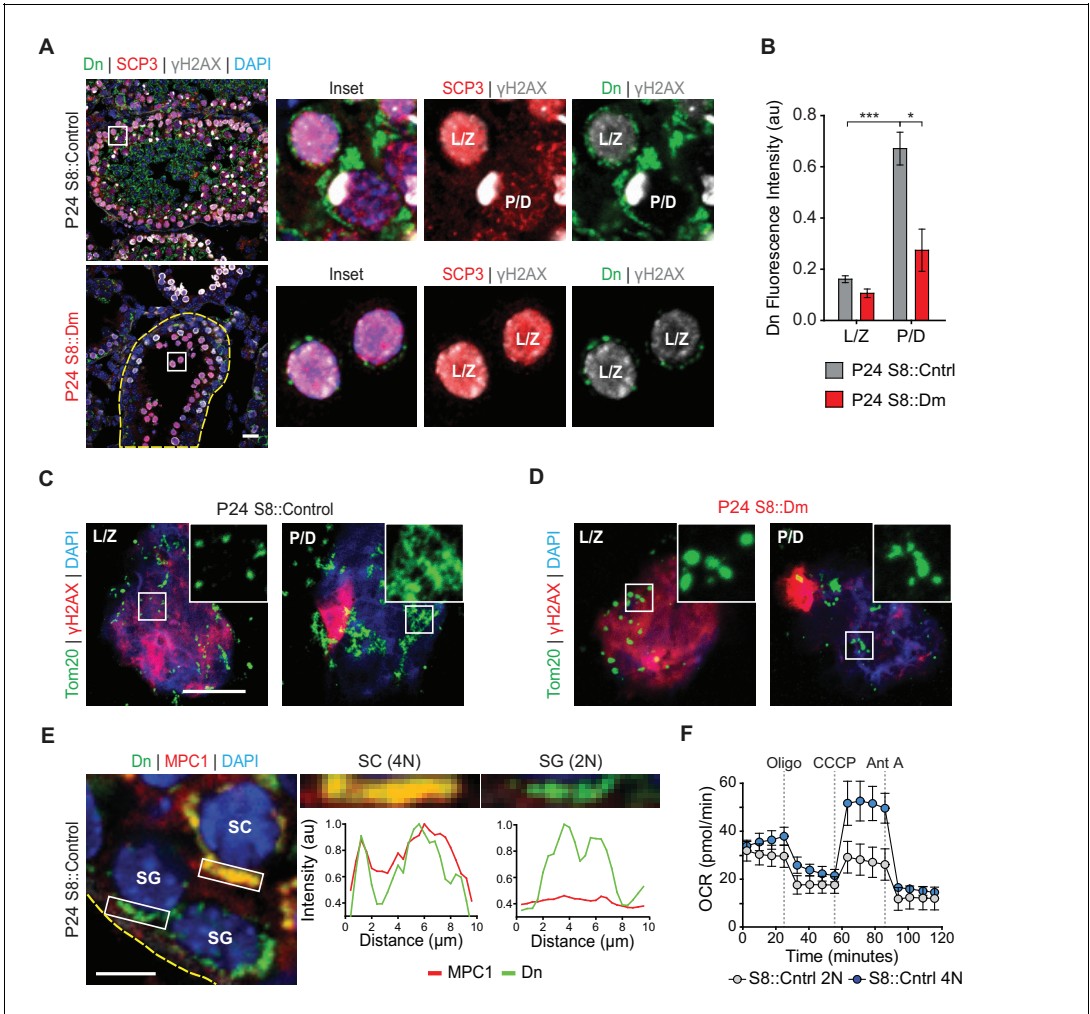

**Figure 3.** Spermatocytes undergo a metabolic shift during meiosis. (**A**) Analysis of mitochondrial content in MPI spermatocytes from testis sections. Mitochondria were visualized by mito-Dendra2 (Dn). γH2AX staining was used to distinguish pachytene/diplotene spermatocytes from leptotene/zygotene spermatocytes, as in *Figure 2C*. For clarity, the border of the mutant seminiferous tubule is outlined by a dashed line. Scale bar, 20 μm. (**B**) Quantification of mito-Dendra2 fluorescence in spermatocytes in testis sections. N = 4. (**C**) Analysis of mitochondrial morphology from meiotic spreads in WT spermatocytes. Mitochondria were visualized with the outer mitochondrial membrane marker, Tom20, and the stages of meiosis were distinguished by γH2AX staining. Note the clustering and elongation in Pachytene/Diplotene spermatocytes. Scale bar, 10 μm. (**D**) Analysis of mitochondrial morphology in S8::Dm spermatocytes. Note the fragmented mitochondrial morphology in Pachytene/Diplotene. Scale bar, 10 μm. (**E**) Selective expression of the mitochondrial pyruvate carrier, MPC1, in WT spermatocytes. The basement membrane at the tubule periphery is indicated by the yellow dashed line. The fluorescence of mitochondria within the boxed region were analyzed by line scanning (right). Arbitrary units, au. Scale bar, 10 μm. (**F**) Comparison of oxygen consumption rates (OCR) from diploid versus tetraploid germ cells from WT adult testes (2–3 months). Germ cells were isolated by FACS and oxygen consumption analyzed with a Seahorse extracellular flux analyzer. 2N, diploid; 4N, tetraploid; Oligo, Oligomycin; CCCP, Carbonyl cyanide m-chlorophenyl hydrazine; Ant A, Antimycin A. The average values from three separate experiments are plotted. SC, spermatocyte; P/D, pachytene/diplotene; L/Z, leptotene/zygotene; SG, spermatogonium. All data are from P24 mice unless otherwise noted. All data are from P24 mice. Data are represented as mean ± SEM. ***p≤0.001; *p≤0.05. For statistical tests used, see Material and methods section.

The online version of this article includes the following figure supplement(s) for figure 3:

**Figure supplement 1.** Analysis of sorted germ cells by immunostaining with specific markers.

**Figure supplement 2.** Quantification of OXPHOS parameters from *Figure 3F*.

seminiferous tubules, confirmed their identities by immunostaining (*Figure 3—figure supplement 1*), and measured their OXPHOS activities using a Seahorse extracellular flux analyzer (*Figure 3F*). Indeed, compared with diploid spermatogonia, tetraploid spermatocytes have substantially greater maximal respiration and spare respiratory capacity, and a modest increase in basal respiration and ATP production (*Figure 3F* and *Figure 3—figure supplement 2*). Taken together, our findings indicate that mitochondrial fusion is required for a metabolic shift during meiosis.

## Long-term mitofusin loss results in depletion of all differentiated germ cell types

To examine the long-term effects of mitofusin loss, we analyzed adult mice at P56. By this time, the testes have undergone repeated rounds of spermatogenesis and should have more complete depletion of mitofusins. PAS-stained testis sections from P56 S8::Dm mice showed greater germ cell depletion and more widespread Sertoli cell vacuolization (*Figure 4A*) compared with P24 mutants. This severe depletion of germ cells was confirmed by flow cytometry, which showed an 80% loss of mito-Dendra2-positive cells from dissociated S8::Dm testes compared with control (*Figure 4B and C*). To identify the stage of the arrest in P56 mice, we quantified all germ cell types using molecular markers and found a depletion of all differentiating germ cell types in S8::Dm mice (*Figure 4D–F* and *Figure 4—figure supplement 1*). Whereas WT seminiferous tubules invariably contain spermatids at this age, nearly half of the P56 mutant tubules (48 ± 7%) contain only spermatogonia (*Figure 4D and E*, and *Figure 4—figure supplement 1*). This histological phenotype is quite distinct from P24 mutants, which contain many tubules arrested in meiosis (*Figure 2B–E*). Consistent with the flow cytometry data, quantification using molecular markers showed that S8::Dm tubules, compared with WT, had greater than 80% depletion of all differentiating germ cell types (*Figure 4F*).

Despite this severe depletion of germ cells, mutant seminiferous tubules retained an outer rim of mito-Dendra2-positive germ cells that stained with PLZF, a marker of undifferentiated spermatogonia (*Figure 4A and D* rightmost panel, and 4F). Because undifferentiated spermatogonia are a heterogeneous population that include stem and progenitor spermatogonia (*de Rooij, 2017*; *Lord and Oatley, 2017*), we sought to better define the identity of spermatogonia retained at the tubule periphery of S8::Dm mice. To this end, we co-labeled testis sections with the pan-undifferentiated spermatogonial marker, PLZF, and the well-established stem-like spermatogonial marker, GFRα1 (*Hara et al., 2014*) (*Figure 4F and G*). Not only were GFRα1-positive spermatogonia present in S8::Dm mice, their numbers were increased over 50%. Because endogenous Stra8 is expressed in late undifferentiated spermatogonia (*Endo et al., 2017*; *Hara et al., 2014*; *Teletin et al., 2019*), we characterized expression of the *Stra8-Cre* driver, which utilizes a 1.4 Kb Stra8 promoter fragment (*Oulad-Abdelghani et al., 1996*; *Zhou et al., 2008*) (*Figure 1—figure supplements 1* and *2*). Quantification from testis sections in WT and S8::Dm mice shows that the vast majority of GFRα1-positive spermatogonia express *Stra8-Cre/Dn* (*Figure 4G* and *Figure 4—figure supplement 2*). Furthermore, over 40% of these GFRα1-expressing spermatogonia are positive for the proliferation marker Ki-67, similar to that in WT animals (*Figure 4H and I*). Thus, mitochondrial fusion is required for maintenance of all differentiated germ cell types but dispensable for self-renewal of stem-like undifferentiated spermatogonia (*Figure 4J*).

To examine whether undifferentiated spermatogonia deplete with age, we examined mice at 4 months of age (P116). Even in these older mice, mutant seminiferous tubules retain an outer rim of undifferentiated spermatogonia that stain with Ki-67 (*Figure 4—figure supplement 3*). To determine whether removal of either mitofusin alone is sufficient to cause depletion of differentiating spermatogonia, we performed histology in S8::Mfn1 and S8::Mfn2 mice (*Figure 4—figure supplement 4*). As in S8::Dm mice, loss of either *Mfn1* or *Mfn2* alone causes severe depletion of germ cells. Quantification of c-Kit-expressing spermatogonia showed that both mutants have severe depletion of differentiating spermatogonia, close to the level observed in S8::Dm mice. Thus, both *Mfn1* and *Mfn2* are required for maintenance of differentiating spermatogonia. Taken together, our data indicate that long-term mitofusin loss impairs maintenance of all differentiated germ cell types, but not self-renewal of stem-like spermatogonia.

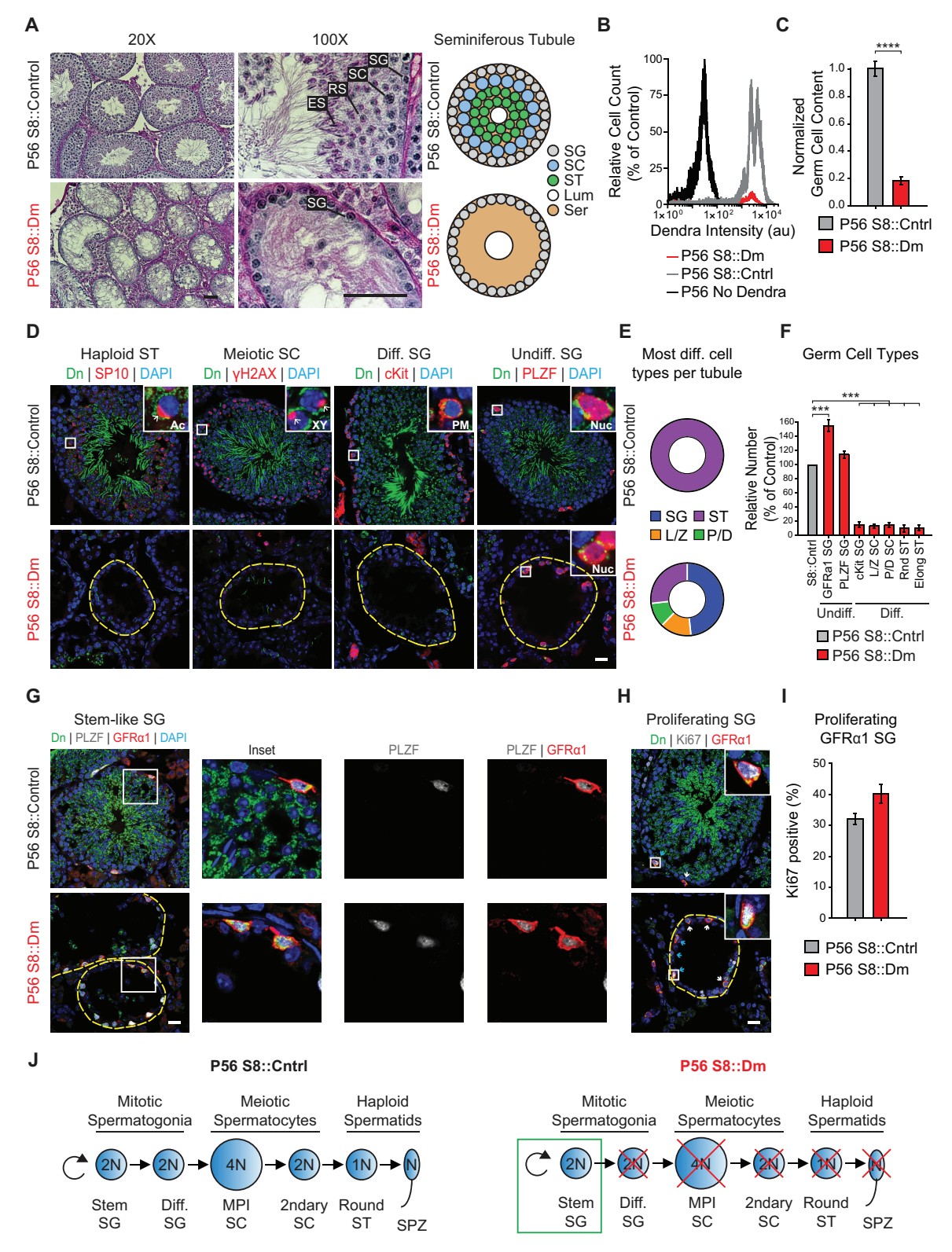

**Figure 4.** Long-term mitofusin loss results in depletion of all differentiated germ cell types. (**A**) PAS histology in P56 testis sections. Scale bar, 50 μm. On the right are stylized depictions of the 100X panels. SG, spermatogonium; SC, spermatocyte; ST, spermatid, Lum, lumen; Ser, Sertoli cell cytoplasm. (**B**) FACS analysis of germ cells in WT and mutant testes. Testes were dissociated and FACS was used to quantify the number of mito-Dendra2-positive germ cells. Dendra fluorescence (x-axis) is shown in arbitrary units (au). The histograms were normalized by plotting the same number of somatic (mito-

*Figure 4 continued on next page*

*Figure 4 continued*

Dendra2-negative) cells between S8::Control and S8::Dm samples. (C) Quantification of the results from (B). N = 5. (D) Immunolabeling of the major germ cell types in WT and mutant animals. Scale bar, 20 μm. (E) Doughnut graphs indicating germ cell types in seminiferous tubule cross sections. Each seminiferous tubule cross section was scored for its most advanced germ cell type. Note that in WT tubules, all seminiferous tubules contain spermatids. For p-values, see *Figure 4—figure supplement 1*. N = 4. (F) Quantification of germ cell types per tubule cross section. Mutant values are plotted relative to control, which is set to 100% and indicated by the gray bar. N = 4. (G) Immunolabeling of stem-like spermatogonia. PLZF marks all undifferentiated spermatogonia and GFRα1 marks stem-like spermatogonia. Note that all GFRα1 spermatogonia express PLZF, but not vice versa. Scale bar, 20 μm. (H) Immunolabeling of the proliferation marker Ki-67 in GFRα1-expressing spermatogonia in testis section. Scale bar, 20 μm. (I) Percent of GFRα1-expressing spermatogonia that are positive for the proliferation marker Ki67. N = 4. (J) Schematic of normal spermatogenesis (left) and the defect in P56 S8::Dm mice (right). SPZ, spermatozoon; ST, spermatid; MPI; Meiotic Prophase I; SC, spermatocyte; P/D, pachytene/diplotene; L/Z, leptotene/zygotene; SG, spermatogonium. Ac, acrosome, XY, sex body; PM, plasma membrane; Nuc, nucleus. N ≥ 4, unless otherwise indicated. Data are represented as mean ± SEM. ****p≤0.0001; ***p≤0.001. For statistical tests used, see the Materials and methods section.

The online version of this article includes the following figure supplement(s) for figure 4:

**Figure supplement 1.** Plot of the most differentiated cell type found in seminiferous tubules.
**Figure supplement 2.** Quantification of *Stra8-Cre/Dn* expression in stem-like GFRα1-expressing spermatogonia.
**Figure supplement 3.** Testis sections from 4-months-old S8::Dm mice stained with PLZF or Ki-67.
**Figure supplement 4.** Both Mfn1 and Mfn2 are required for spermatogonial differentiation.

## Mitofusin-deficient germ cells have aberrant cristae ultrastructure and increased apoptosis

We next visualized mitochondrial ultrastructure in testis sections via 3D EM tomography (*Figure 5A–D*, and *Figure 5—figure supplement 1*). Mutant mitochondria had aberrant cristae morphology characterized by a fourfold reduction in the number of linear cristae elements and a fourfold increase in swollen, vesiculated cristae (*Figure 5A and B*). Despite their isolated appearance, all vesiculated cristae in S8::Dm mitochondria could be traced back to the inner mitochondrial membrane using 3D EM tomography (*Video 4*). Furthermore, many cristae junctions (CJs) in mutant mitochondria are significantly widened in diameter (*Figure 5C and D*, and *Figure 5—figure supplement 1*). While most of the CJs in control spermatocytes are narrower than 22.5 nm (64%), the majority in S8::Dm are wider (78%) (*Figure 5—figure supplement 1*). In fact, 50% of mutant CJs, compared to 13% in WT, are wider than 27.5 nm. The aberrant cristae morphology and increased CJ diameter are suggestive of increased mitochondrial apoptosis in S8::Dm mice. To verify this, we performed terminal deoxynucleotidyl transferase dUTP nick end labeling (TUNEL) in testis sections. Indeed, S8::Dm sections show increased TUNEL-positive cells, indicative of cell death by apoptosis (*Figure 5E and F*).

## Mitofusin-deficient MEFs have reduced OXPHOS subunits and mitochondrial ribosomes

To understand the cellular mechanism for the spermatogenic and mitochondrial defects, we sought for clues in mitofusin-null mouse embryonic fibroblasts (MEFs), where we could readily obtain sufficient material to do proteomic analysis. We used stable isotope labeling of amino acids in cell culture (SILAC) to quantitatively compare the mitochondrial proteome of *Mfn1/Mfn2*-null MEFs with WT controls (*Figure 6* and *Figure 6—figure supplement 1*). Gene Ontology (GO) analysis revealed two general categories of proteins that are significantly reduced in mitochondria from *Mfn1/Mfn2*-null MEFs: the mitochondrial respiratory chain and the mitochondrial ribosome (*Figure 6A* and *Supplementary file 1*). Indeed, 31 out of 34 (91%) identified complex I subunits, all 12 (100%) complex IV subunits, and 74 out of 80 (93%) mitochondrial ribosomal subunits were reduced in *Mfn1/Mfn2*-null MEFs (*Supplementary file 2*). Of the proteins that reached statistical significance (p<0.05), all 14 complex I subunits, all four complex IV subunits, and all 18 mitochondrial ribosomal subunits were reduced (*Figure 6B*). Consistent with reduced OXPHOS activity, *Mfn1/Mfn2*-null MEFs had significantly lower levels of the mitochondrial pyruvate carrier, MPC2 (*Bricker et al., 2012*; *Vanderperre et al., 2016*). Additionally, TFAM, a mitochondrial transcriptional activator and mtDNA packaging protein, was significantly reduced. Intriguingly, 14 out of 18 (78%) complex I and IV assembly factors were increased in *Mfn1/Mfn2*-null MEFs and of the seven that reached statistical significance, all were increased (*Figure 6B*). Our GO analysis also revealed an upregulation of mitochondrial import proteins in *Mfn1/Mfn2*-null MEFs (*Figure 6A*). 17 out of 21 (81%) detected mitochondrial import proteins were increased, and of the 10 that reached statistical significance, all were

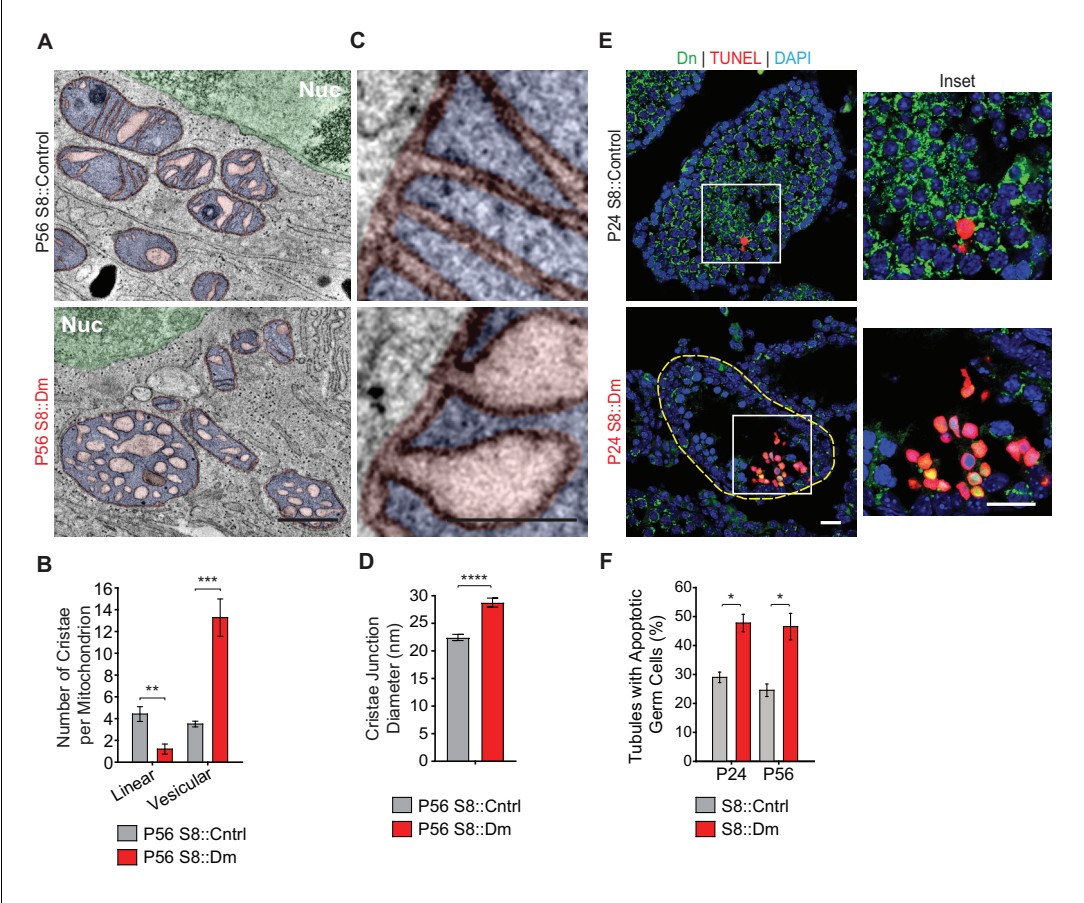

**Figure 5.** Mitofusin-deficient germ cells have increased apoptosis. (**A**) EM tomograms from adult P56 testis sections highlighting mitochondrial cristae morphology. The following pseudocolors are used: mitochondrial matrix, blue; cristae lumen, pink; nucleus, green. Nuc, nucleus. (**B**) Quantification of mitochondrial cristae morphology from EM tomograms. At least 70 cristae junctions were quantified from at least four different EM tomograms from a single mouse from each genotype. (**C**) EM tomograms from testis sections highlighting cristae junctions. Mitochondrial matrices are colored blue, and mitochondrial intermembrane spaces and cristae are colored pink. (**D**) Quantification of cristae junction diameters from EM tomograms. (**E**) TUNEL staining of PFA-fixed testis sections. Cells were labeled for TUNEL, mito-Dendra2 (Dn), and nuclei (DAPI). Scale bars, 20 µm. (**F**) Quantification showing the percentage of seminiferous tubule cross-sections containing TUNEL-positive cells. N = 3. Data are represented as mean ± SEM. ****p≤0.0001; ***p≤0.001; **p≤0.01; *p≤0.05. For statistical tests used, see the Materials and methods section.

The online version of this article includes the following figure supplement(s) for figure 5:

**Figure supplement 1.** Size distribution of cristae junction diameters in spermatocytes from *Figure 5C and D*.

increased in *Mfn1/Mfn2*-null MEFs (*Figure 6B*). Taken together, these data suggest that the reduced OXPHOS activity in *Mfn1/Mfn2*-null MEFs (*Chen et al., 2005*) is caused by downregulation of OXPHOS components and mitochondrial ribosomal subunits.

## Mitofusin knockdown in immortalized spermatocytes causes reduced OXPHOS subunits and activity

With the proteomic insights from MEFs, we examined whether OXPHOS deficiency also occurs in germ cells upon loss of mitofusins. To this end, we utilized the widely used spermatocyte cell line, GC-2spd(ts), referred to herein as GC (*Figure 7* and *Figure 7—figure supplements 1–3*) (*Hofmann et al., 1994*; *Hofmann et al., 1995*; *Rahman and Huhtaniemi, 2004*). To block mitochondrial fusion in GCs, we constructed an shRNA vector for dual Mfn1 and Mfn2 knockdown (shMfn1;Mfn2). Effective mitofusin knockdown was verified by western blotting (*Figure 7—figure supplement 1*) and by quantifying mitochondrial morphology (*Figure 7A and B*). Strikingly, mitofusin knockdown in GCs causes widespread mitochondrial fragmentation and reduction in all OXPHOS

subunits tested by western blotting: complex I subunit (NdufB8), complex II subunit SDHB, complex III subunit UQRC2, complex IV subunit MTCOI, and complex V subunit ATP5A (*Figure 7C and D*). To measure OXPHOS activity directly, we measured oxygen consumption in GCs using a Seahorse extracellular flux analyzer. Mitofusin knockdown caused a severe reduction in basal and maximal respiration, ATP production, and spare capacity (*Figure 7E and F*), but not glycolysis (*Figure 7G*). Interestingly, mitofusin knockdown in MEFs caused mitochondrial fragmentation (*Figure 7—figure supplement 2*), but not OXPHOS deficiency (*Figure 7—figure supplement 3*). Our knockdown procedure, while effective, was not efficient enough to reproduce the OXPHOS deficiency that characterizes mitofusin-null MEFs (*Chen et al., 2005*). Taken together, these data indicate that GCs are particularly sensitive to loss of mitofusins.

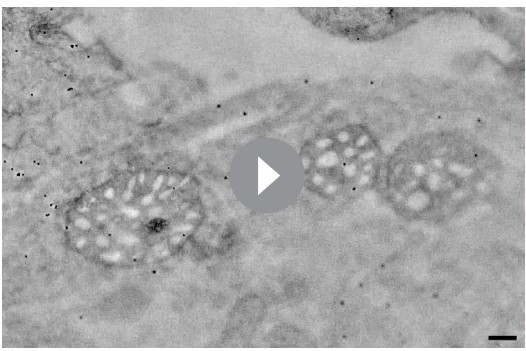

**Video 4.** EM tomogram of mitochondrion from P56 S8::Dm testis section showing that all cristae in the 400 nm testis section that can be traced back to the mitochondrial inner membrane are connected via cristae junctions. Scale bar 200 nm.
https://elifesciences.org/articles/51601#video4

## Mitofusin-deficient spermatocytes have reduced OXPHOS components and activity in vivo

To test these findings in germ cells in vivo, we performed immunofluorescence against various OXPHOS components in spermatocytes in testis sections (*Figure 8A and B*). In WT mice, spermatocytes have strong and homogeneous expression of OXPHOS components that uniformly overlaps with mito-Dendra2-labeled mitochondria. In contrast, most spermatocytes in S8::Dm mice show a heterogeneous staining pattern in which a substantial subset of mito-Dendra2-positive mitochondria clearly lack OXPHOS components. Consistent with the proteomics in cultured cells, S8::Dm spermatocytes have reduced and heterogeneous expression of the complex I subunit NdufB6, the complex IV subunit MTCOI, and the mitochondrial pyruvate carrier MPC1 (an obligate binding partner of MPC2). We noted that the mtDNA-encoded MTCOI has the most heterogeneous expression. Similarly, many shMfn1;Mfn2 GCs display reduced or heterogeneous MTCOI expression, with a substantial subset of mitochondria lacking MTCOI (*Figure 8—figure supplement 1*).

In agreement with the progressive loss of differentiated spermatogonia in S8::Dm mice, there is also mitochondrial heterogeneity in this cell type (*Figure 8—figure supplement 2*). Surprisingly, we also find mitochondrial heterogeneity in undifferentiated spermatogonia, which are not depleted in S8::Dm mice. Thus, loss of mitofusins leads to heterogeneous mitochondria in all germ cell types. However, the undifferentiated spermatogonia can apparently tolerate this defect, suggesting that these cells are physiologically different.

To test for functional changes in mitochondrial activity, we performed COX/SDH enzyme histochemistry on spermatozoa isolated from S8::Mfn1 and S8::Mfn2 mice and found reduction of both COX and SDH activities (*Figure 8—figure supplement 3*). Because S8::Dm mice do not produce any sperm, we performed COX/SDH enzyme histochemistry in fresh-frozen testis sections and found a severe reduction of COX and SDH activities in S8::Dm germ cells (*Figure 8C*). Thus, consistent with data in MEFs and immortalized spermatocytes, mitofusin depletion in germ cells in vivo causes ultrastructural defects, heterogeneity in OXPHOS expression, and reduced OXPHOS subunits and activity (*Figure 9*).

## Discussion

Mfn1 is known to be required in spermatocytes in the mouse (*Zhang et al., 2016*), but it was unclear whether this requirement simply reflected the need for basal mitochondrial dynamics as a housekeeping function. Our study reveals that spermatocytes require mitochondrial fusion as they undergo an acute upregulation of OXPHOS during MPI (*Figure 9*). Analysis of the single mutant mice showed that *Mfn1* but not *Mfn2* is essential for this transition. We show that this transition coincides with

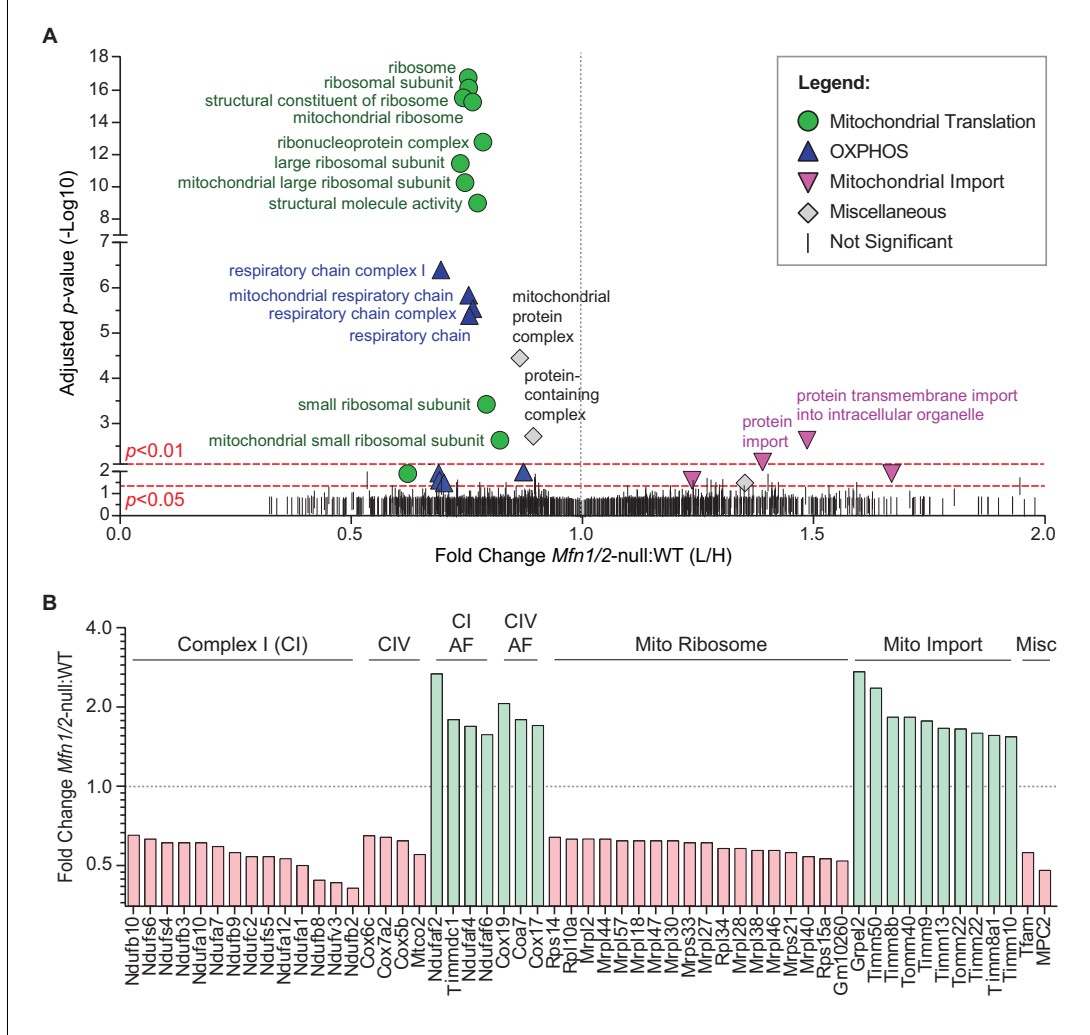

**Figure 6.** Mitofusin-deficient MEFs have reduced OXPHOS subunits and mitochondrial ribosomes. (**A**) Volcano plot showing enriched Gene Ontology terms from SILAC experiments in MEFs. SILAC was performed on isolated mitochondria from WT and *Mfn1/Mfn2*-null MEFs. See also *Supplementary file 1*. (**B**) Comparison of individual mitochondrial proteins from SILAC analysis of MEFs. Bar charts show the protein ratios in *Mfn1/Mfn2-null* versus WT. Only proteins with ratios that reached statistical significance (p≤0.05) are plotted. For a complete list of proteins, see *Supplementary file 1*. The black dashed line indicates a mutant to WT ratio of 1:1. Magenta bars indicate proteins that are reduced and green bars indicate proteins that are increased. See also *Supplementary file 2*. Two biological replicates were used. Assembly factor, AF. For statistical tests used, see the Materials and methods section.

The online version of this article includes the following figure supplement(s) for figure 6:

**Figure supplement 1.** Schematic of SILAC experiment in MEFs.

increased mitochondrial density, mitochondrial elongation, and upregulation of OXPHOS activity, which are all blocked in fusion-deficient spermatocytes. The increased OXPHOS in pachytene spermatocytes is likely required to fuel ATP-consuming biochemical reactions involved in homologous recombination and DSB repair. Consistent with this idea, the metabolic environment of spermatocytes changes as they initiate MPI and traverse the blood-testis barrier. In this isolated compartment, spermatocytes are segregated from the vasculature and interstitial fluid and thus rely on metabolites secreted by the nursing Sertoli cells (*Boussouar and Benahmed, 2004*; *Stanton, 2016*). As a result, spermatocytes switch their energy dependence from glucose to lactate and pyruvate, which are preferred substrates for OXPHOS (*Bajpai et al., 1998*; *Grootegoed et al., 1984*; *Nakamura et al., 1984*; *Rato et al., 2012*). In agreement with upregulation of OXPHOS during pachytene, others and we have found high expression of the mtDNA-encoded COXI protein (*Figure 8A*, middle panel)

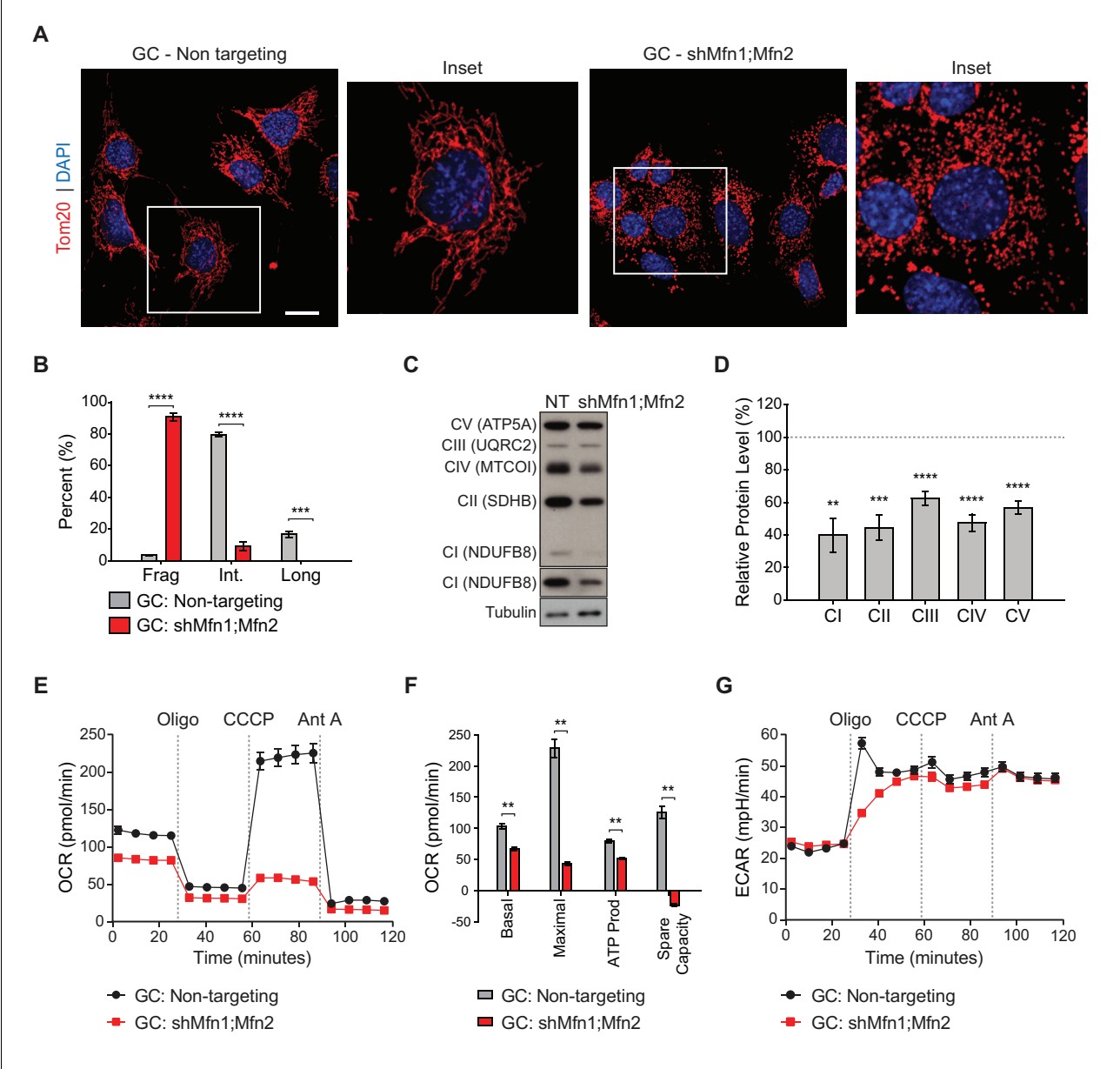

**Figure 7.** Mitofusin-deficient immortalized spermatocytes have reduced OXPHOS subunits and activity. (**A**) Immunostaining against the mitochondrial outer membrane marker, Tom20 in an immortalized spermatocyte cell line, GC-2Spd(ts) (GC). Note the mitochondrial fragmentation in shMfn1;Mfn2 treated cells compared to the non-targeting control. Scale bar, 20 μm. (**B**) Quantification of mitochondrial morphology in GCs. Mean values from three replicates are plotted. (**C**) Western blots showing various respiratory chain complexes in GCs. NT, non-targeting. (**D**) Densitometry analysis of western blots. Mean values from four replicates are plotted. (**E**) Oxygen Consumption Rate (OCR) measurements in GCs. (**F**) Quantification from E. (**G**) Extracellular Acidification Rate (ECAR) measurements in GCs. Oligo, oligomycin; CCCP, carbonyl cyanide m-chlorophenyl hydrazine; Ant A, antimycin A. Data are represented as mean ± SEM. ****p≤0.0001; ***p≤0.001; **p≤0.01. For statistical tests used, see the Materials and methods section. The online version of this article includes the following figure supplement(s) for figure 7:

**Figure supplement 1.** Mitofusin knockdown in immortalized spermatocytes.
**Figure supplement 2.** Quantification of mitochondrial morphology in MEFs.
**Figure supplement 3.** Oxygen Consumption Rate.

(*Jiang et al., 2017*) and mRNA (*Saunders et al., 1993*) in pachytene spermatocytes. Furthermore, we find that spermatocytes, compared with pre-meiotic spermatogonia, switch on expression of the

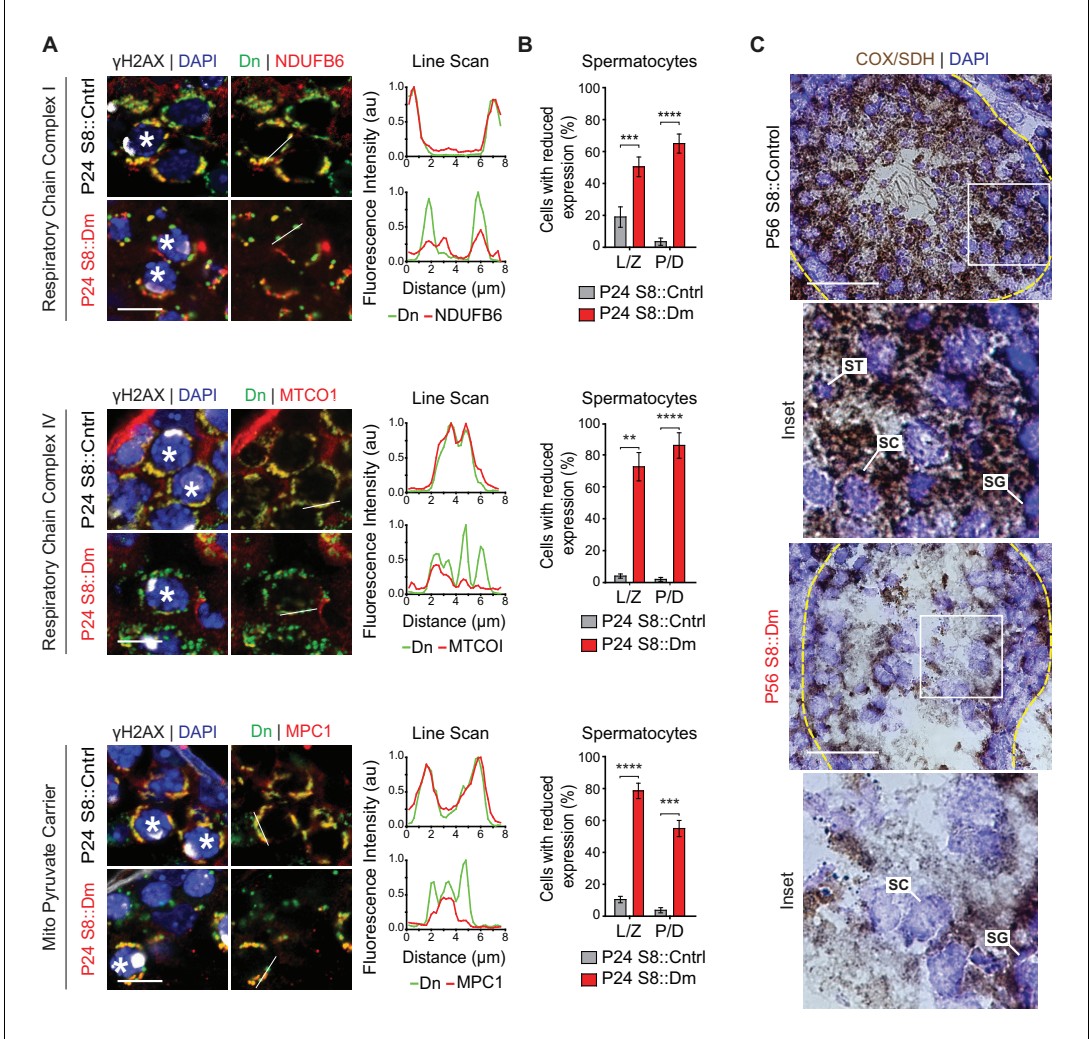

**Figure 8.** In vivo spermatocytes have heterogeneous mitochondria and reduced OXPHOS activity. (**A**) Immunolabeling of NDUFB6, MTCOI, and MPC1 in spermatocytes in testis sections. Line scans are shown as arbitrary units (au) to the right. Scale bars, 20 μm. (**B**) Quantification showing the percentage of spermatocytes with reduced or heterogeneous staining. L/Z, leptotene/zygotene; P/D, pachytene/diplotene. N = 4. (**C**) COX/SDH enzyme histochemistry in adult testis sections. Note that mutant sections have much reduced OXPHOS activity. Scale bars, 50 μm. ST, spermatid; SC, spermatocyte; SG, spermatogonium. Data are represented as mean ± SEM. ****p≤0.0001; ***p≤0.001; **p≤0.01. For statistical tests used, see the Materials and methods section.

The online version of this article includes the following figure supplement(s) for figure 8:

**Figure supplement 1.** Immunolabeling of the mtDNA-encoded respiratory chain complex IV subunit MTCOI (green).

**Figure supplement 2.** Mitochondrial heterogeneity in spermatogonia.

**Figure supplement 3.** OXPHOS deficiency in Mfn1 and Mfn2 null sperm.

mitochondrial pyruvate carrier, MPC1, presumably to accommodate the greater flux of pyruvate into the OXPHOS pathway. Changes in pyruvate metabolism, driven by MPC expression, have been shown to regulate differentiation of stem cells in other systems (*Flores et al., 2017*; *Schell et al., 2017*).

Our findings are in agreement with previous indications that mitochondrial morphology, cristae structure, and OXPHOS activity change during MPI. Round spherical mitochondria with orthodox cristae in early MPI convert in pachytene to more elongated mitochondria with condensed cristae (*De Martino et al., 1979*; *Meinhardt, 1999*; *Seitz et al., 1995*), an ultrastructure associated with active OXPHOS (*Hackenbrock, 1966*; *Mannella, 2006*). Furthermore, mitochondrial function has been shown to be important for this particular developmental stage. Arrest in early MPI occurs in

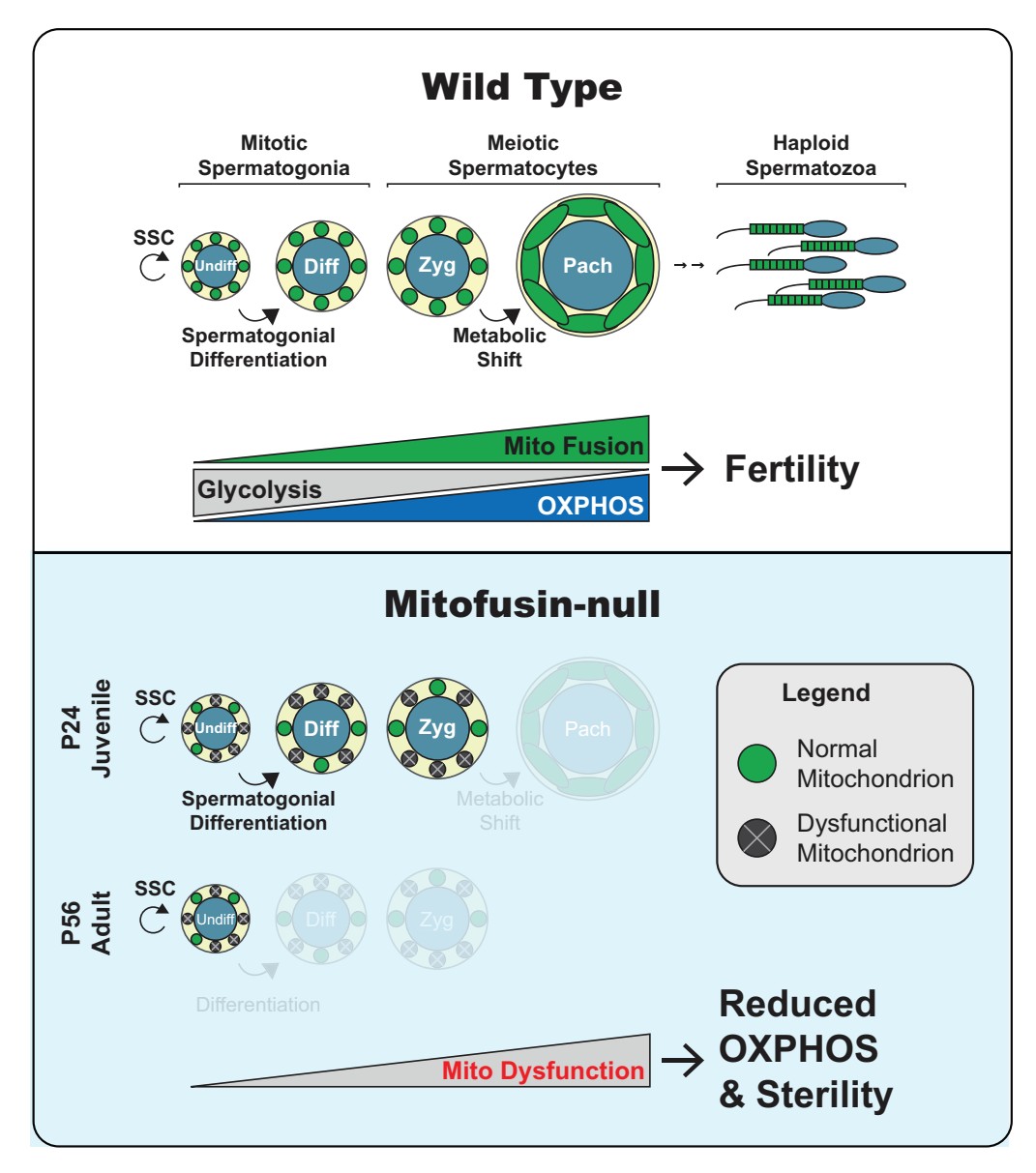

**Figure 9.** Model. Top panel. Two key transitions during normal spermatogenesis are indicated: spermatogonial differentiation and the metabolic shift during meiosis. The metabolic shift includes increased mitochondrial biogenesis, fusion, and OXPHOS. Bottom panel. In juvenile mitochondrial fusion-deficient mice, spermatocytes have functionally heterogeneous mitochondria that are unable to accomplish this metabolic shift, resulting in meiotic arrest. Adult mutant mice have an earlier defect, wherein all differentiating germ cell types are lost but self-renewal of stem-like undifferentiated spermatogonia remains intact.

mouse models with a pathogenic mtDNA deletion (*Nakada et al., 2006*) or inability to utilize mitochondrial ATP (*Brower et al., 2009*). In addition, mouse models that accumulate mtDNA mutations exhibit male infertility (*Jiang et al., 2017*; *Kujoth et al., 2005*; *Trifunovic et al., 2004*).

In contrast to the Mfn1 study (*Zhang et al., 2016*), we show that mitofusins are required not only in spermatocytes, but also in differentiating spermatogonia (*Figure 9*). S8::Mfn1, S8::Mfn2, and S8::Dm mice exhibit a progressive loss of all differentiated germ cell types, including the differentiating spermatogonia that give rise to spermatocytes. The age-dependent loss is likely caused by the progressive depletion of mitofusin gene products due to the iterative nature of spermatogenesis. At P24, during the first round of spermatogenesis, germ cells may contain residual levels of mitofusin activity sufficient to mitigate effects on spermatogonia. As a result, the major defect is found during

MPI, which is more sensitive to ATP depletion because of the metabolic shift described above. With long-term mitofusin loss in adult mice, even the less sensitive differentiating spermatogonia are affected. Mitochondrial fusion, however, is dispensable for self-renewal of undifferentiated spermatogonia, even though these cells display mitochondrial heterogeneity with loss of mitofusins. Undifferentiated spermatogonia, differentiating spermatogonia, and spermatocytes therefore show progressively increasing requirements for mitochondrial fusion. These results are consistent with a paradigm in various (but not all) stem cell systems, wherein stem cells compared with differentiated cells are less dependent on mitochondrial OXPHOS (*Chen and Chan, 2017*).

The two major phenotypes of S8::Dm mice—meiotic failure and spermatogonial differentiation—are also observed with loss of *Mfn1* alone. This observation suggests that these defects should be attributed to loss of mitochondrial fusion, rather than non-fusion functions, like ER-mitochondria tethering, that have been attributed to Mfn2 (*de Brito and Scorrano, 2008*). To further test that our spermatogenesis defects are due to loss of mitochondrial fusion, it will be helpful to analyze mice lacking other mitochondrial dynamics factors, particularly *Opa1*.

To elucidate the cellular mechanism for the spermatogenic arrest we performed proteomics in MEFs and found downregulation of respiratory chain complex I and IV subunits, mitochondrial translation proteins, and the mitochondrial pyruvate carrier, MPC2. A previous proteomic study found that *Mfn2*-null MEFs have a defect in the coenzyme Q pathway, but no change in the levels of respiratory chain complexes (*Mourier et al., 2015*). In our study, full inhibition of mitochondrial fusion by deletion of both mitofusins uncovers their role in maintaining the respiratory chain and mitochondrial translation. Finally, we find an increase in the respiratory chain complex I and IV assembly factors as well as mitochondrial import proteins in *Mfn1/Mfn2*-null MEFs. These increases may be indicative of a compensatory stress response to the cellular dysfunction caused by mitochondrial fusion deficiency. Importantly, the metabolic derangements revealed by proteomics in MEFs extend to a spermatocyte cell line and to spermatocytes in vivo. Thus, our findings provide a rationale for how mitochondrial fusion safeguards mitochondrial function and male fertility.

## Materials and methods

### Generation of S8::Dm mice and characterization of Stra8-Cre expression

All mouse experiments were approved by the California Institute of Technology (Caltech) Institutional Animal Care and Use Committee. S8::Control (Stra8-Cre$^{+/tg}$; $Mfn1^{+/loxP}$; $Mfn2^{+/loxP}$; $Rosa26^{P-hAM(+/loxP)}$), S8::Mfn1 (Stra8-Cre$^{+/tg}$; $Mfn1^{\Delta/loxP}$; $Mfn2^{+/lox}$; $Rosa26^{PhAM(+/loxP)}$), S8::Mfn2 (Stra8-Cre$^{+/tg}$; $Mfn1^{+/loxP}$; $Mfn^{\Delta/loxP}$; $Rosa26^{PhAM(+/loxP)}$), and S8::Dm (Stra8-Cre$^{+/tg}$; $Mfn1^{\Delta/loxP}$; $Mfn2^{\Delta/loxP}$; $Rosa26^{PhAM(+/loxP)}$) mice were generated by crossing Stra8-Cre; $Mfn1^{+/\Delta}$; $Mfn2^{+/\Delta}$ mice to $Mfn1^{loxP/loxP}$; $Mfn2^{loxP/loxP}$; $Rosa26^{PhAM(loxP/loxP)}$ mice. The null alleles $Mfn1^{\Delta}$ and $Mfn2^{\Delta}$ (*Chen et al., 2003*), the conditional alleles $Mfn1^{loxP}$ and $Mfn2^{loxP}$ (*Chen et al., 2007*), the Stra8-Cre driver (Jackson Laboratory #017490) (*Sadate-Ngatchou et al., 2008*), and the $Rosa26^{PhAM}$ allele containing mito-Dendra2 (*Pham et al., 2012*) were all described previously. All mice were maintained on a C57B6 background except for $Mfn1^{loxP/loxP}$; $Mfn2^{loxP/loxP}$; $Rosa26^{PhAM(loxP/loxP)}$ mice, which were maintained on a C57B6/129S mixed genetic background.

### Epididymal sperm counting and analysis of morphology and motility

Mice were euthanized at P56, and epididymides were dissected and thoroughly minced in 1.7 ml microcentrifuge tubes containing 1 ml PBS. Samples were incubated at 37°C for 20 min to allow sperm to swim out. 900 µl of the supernatant was transferred into a fresh microcentrifuge tube. For sperm counting and morphology analysis, samples were allowed to settle for several hours for sperm to stop swimming before counting on a hemocytometer. Sperm counts were normalized to the weight of the epididymides of each mouse. For motility quantification, freshly isolated sperm were transferred to a hemocytometer, and the proportion of motile sperm was quantified. For motility movies, freshly isolated sperm were transferred to glass-bottom FluoroDish Cell Culture Plates (FD35-100) and imaged at one frame per second for 10 s using a confocal microscope as described below.

## Periodic acid-Schiff staining

After dissection, testes were fixed in Bouin's fixative overnight at 4°C, dehydrated in a 30–90% ethanol gradient, cleared in Xylenes, and embedded in paraffin. Tissue blocks were sectioned at 7 μm, deparaffinized, and rehydrated before staining. Briefly, slides were incubated with 1% periodic acid (Electron Microscopy Sciences (EMS); 19324–10) for 30 min at RT, washed in running water for 5 min, then rinsed in deionized water. Slides were incubated with Schiff's reagent (EMS; 260582–05) for 30 min at RT and washed as described above before counterstaining with Hematoxylin Gill 2 for 30 s at RT. Slides were washed in running water for 1 min, dehydrated with ethanol, cleared with xylene, then mounted using Cytoseal XYL mounting media (Thermo Fisher Scientific; 22-050-262).

## Immunofluorescence

For immunostaining of tissue sections, testes were cut at the poles, fixed in 4% PFA for 4 hr at 4°C, incubated with 30% sucrose in PBS overnight at 4°C (or until tissues sank), incubated in a 1:1 solution of 30% sucrose in PBS and optimal cutting temperature (OCT) embedding medium (Thermo Fisher Scientific; NC9636948) for 15–30 min, then embedded in OCT medium and frozen in dry ice. Tissue blocks were sectioned at 10 μm onto glass slides, dried overnight, and stored at −80°C until ready for immunostaining. Frozen slides were briefly thawed at room temperature, rehydrated in PBS, permeabilized with 0.15% TX-100 for 15 min, and blocked for 1 hr using Blocking Buffer (10% FBS, 3% BSA, 0.05% TX-100 in PBS). Slides were incubated with primary antibodies in a humidified chamber overnight at 4°C, washed three times in PBS for 15 min each, then incubated with secondary antibodies in a humidified chamber for 2.5 hr at RT. Slides were counterstained with DAPI, washed as described above, mounted with Fluorogel (EMS; 17985–10), covered with a coverglass, sealed with nail polish, and stored at 4°C before imaging.

For immunostaining of FACS sorted germ cells, cells were plated onto 8-chamber glass slides (VWR, 62407–296) pre-coated with Cell-Tak (Corning; 354240). After cell adhesion, cells were washed with PBS, fixed in 10% Neutral Buffered Formalin (Sigma Aldrich, HT501128-), and immunostained as described above.

## Western blotting

Cells were grown to confluency in 6-well plates and lysed in 200 ul of Lysis Buffer (20 mM Tris-Cl, 150 mM NaCl, 2 mM EDTA, 1% Triton X-100, pH 7.5 with Halt Protease Inhibitor added to 1X). Proteins (5 μg) were separated by SDS-PAGE and transferred to Polyvinylidene difluoride membranes using semi-dry electrophoresis. Membranes were blocked with 5% non-fat dry milk for 1 hr at RT, followed by primary antibody incubation overnight at 4°C. HRP-conjugated secondary antibodies were applied for 2 hr at RT. Immobilon Western chemiluminescent HRP substrate (Millipore 90720) was added according to the manufacturer's instructions and the protein bands were visualized on X-ray film (Amersham Hyperfilm MP). For analysis, densitometry was performed in ImageJ and protein levels were normalized to that of tubulin.

## Antibodies

The following antibodies were used for immunofluorescence. rabbit anti- γH2AX (ab11174, Abcam); mouse anti-γH2AX (ab26350, Abcam); rabbit anti-c-Kit (3074S, Cell Signaling Technology); rabbit anti-GFRα1; (AF560-SP, R and D Systems); rabbit anti-Ki67 (ab15580, Abcam); rabbit anti-MPC1 (HPA045119, Millipore Sigma); mouse anti-MTCOI (ab14705, Abcam); mouse anti-Ndufb6 (ab110244, Abcam); rabbit anti-PLZF (SC-22831, Santa Cruz Biotech); mouse anti-PLZF (SC-28319, Santa Cruz Biotech); rabbit anti-SCP3; (ab15093, Abcam); mouse anti-SCP3 (ab97672, Abcam); mouse anti-SDHA (ab14715, Abcam); guinea pig anti-SP-10 (Gift from Prabhakara P. Reddi); rabbit anti-Tom20 (SC-11415, Santa Cruz, Biotech); mouse anti-Total OXPHOS (ab110413, Abcam). The following antibodies were used for western blotting. Chicken anti-Mfn1 (*Chen et al., 2003*); rabbit anti-Mfn2 (D2D10S #9482, Cell Signaling Technology); mouse anti-tubulin (T6199, Sigma).

## Chromosome spreading of meiotic spermatocytes

Chromosome spreading was performed as described previously (*Gaysinskaya et al., 2014*; *Peters et al., 1997*). Briefly, testes were decapsulated and placed into 10 cm petri dishes containing 3 ml MEM-α. Seminiferous tubules were loosened under a dissecting microscope with

forceps, and the interstitial cells were washed away. The seminiferous tubules were transferred to a fresh 10 cm petri dish containing 300 μl MEM-α and the tubules were torn between two fine forceps for 3–5 min. A cell suspension was made with an additional 1 ml MEM-α using a P1000 pipette and the large tubular remnants were removed by centrifugation at 500 rpm (27 RCF) for 1 min. The cells were concentrated by centrifugation for 7 min at 1000 rpm (106 RCF), and 500 μl of the supernatant was discarded. Upon re-suspending cells, 600 μl of Hypotonic Buffer was added to 600 μl of the cell suspension and incubated for 7 min at RT. The suspension was spun at 1000 rpm (106 RCF) for 7 min and 1100 μl of the supernatant was discarded. The cells were re-suspended in residual hypotonic buffer (this is an essential step to prevent cell clumping) before adding 1 ml of 100 mM working sucrose solution. 100–200 μl of cell suspension was added to glass slides pre-coated with Fixation Buffer (1% PFA, 0.15% TX-100) and incubated for 1–1.5 hr at RT in a humidified chamber. Slides were stored at −80°C until they were ready for immunostaining. Slides were immunostained as described above. For immunostaining mitochondria, TX-100 was omitted from the Fixation Buffer.

## Electron microscopy and dual-axis tomography

Mouse testes were dissected and immediately fixed with cold 3% glutaraldehyde, 1% paraformaldehyde, 5% sucrose in 0.1 M sodium cacodylate trihydrate. Pre-fixed pieces of tissue were rinsed with fresh cacodylate buffer and placed into brass planchettes (Type A; Ted Pella, Inc, Redding, CA) pre-filled with 10% Ficoll in cacodylate buffer. Samples were covered with the flat side of a Type-B brass planchette and rapidly frozen with a HPM-010 high-pressure freezing machine (Leica Microsystems, Vienna Austria). The frozen samples were transferred under liquid nitrogen to cryotubes (Nunc) containing a frozen solution of 2.5% osmium tetroxide, 0.05% uranyl acetate in acetone. Tubes were loaded into an AFS-2 freeze-substitution machine (Leica Microsystems) and processed at −90°C for 72 hr, warmed over 12 hr to −20°C, held at that temperature for 8 hr, then warmed to 4°C for 2 hr. The fixative was removed, and the samples were rinsed 4x with cold acetone, and then were infiltrated with Epon-Araldite resin (Electron Microscopy Sciences, Port Washington PA) over 48 hr. Samples were flat-embedded between two Teflon-coated glass microscope slides, and the resin polymerized at 60°C for 24–48 hr.

Flat-embedded testis samples were observed with a stereo dissecting microscope, and appropriate regions were extracted with a microsurgical scalpel and glued to the tips of plastic sectioning stubs. Semi-thick (400 nm) serial sections were cut with a UC6 ultramicrotome (Leica Microsystems) using a diamond knife (Diatome, Ltd. Switzerland). Sections were placed on Formvar-coated copper-rhodium slot grids (Electron Microscopy Sciences) and stained with 3% uranyl acetate and lead citrate. Gold beads (10 nm) were placed on both surfaces of the grid to serve as fiducial markers for subsequent image alignment. Grids were placed in a dual-axis tomography holder (Model 2040, E. A. Fischione Instruments, Export PA) and imaged with a Tecnai TF30ST-FEG transmission electron microscope (300 KeV) equipped with a 2k × 2 k CCD camera (XP1000; Gatan, Inc Pleasanton CA). Tomographic tilt-series and large-area montaged overviews were acquired automatically using the SerialEM software package (*Mastronarde, 2005*). For tomography, samples were tilted + /- 64° and images collected at 1° intervals. The grid was then rotated 90° and a similar series taken about the orthogonal axis. Tomographic data was calculated, analyzed and modeled using the IMOD software package (*Kremer et al., 1996*; *Mastronarde, 2008*) on MacPro computers (Apple, Inc, Cupertino, CA).

## Apoptotic cell labeling

To label apoptotic nuclei, the TUNEL assay was performed in PFA-fixed, OCT-embedded testis sections using the ApopTag Red In Situ Apoptosis Detection Kit (Millipore; S7165) according to the manufacturer's protocol. Nuclei were counterstained with DAPI.

## Imaging and image processing

Confocal fluorescence images and videos were acquired using an inverted Zeiss LSM 710 confocal microscope with a 60X Plan-Apochromat objective. Epifluorescence and bright-field images were acquired using an upright Nikon Eclipse Ni-E fluorescence microscope equipped with a Ds-Ri2 camera and CFI Plan Apochromat Lambda objectives. For PAS histology images, Z stacks were acquired and all-in-focus images were created using the NIS Elements Extended Depth of Focus plugin. All

images were processed using ImageJ. All image modifications were performed on entire images (no masking was used) and were performed identically between genotypes.

## Germ cell quantification from testis sections

Germ cells were counted from 10 µm testis sections using the germ cell markers described in the main text. Quantification was restricted to germ cells within round transverse sections of seminiferous tubules and is reported as either the mean number of germ cells per seminiferous tubule cross section or the most differentiated cell type per seminiferous tubule cross section. Only germ cells expressing Dendra2 were included in the quantification to exclude cells without *Stra8-Cre* expression. For each genotype, at least 50 transverse sectioned seminiferous tubules were quantified from at least four mice.

## COX/SDH enzyme histochemistry

COX/SDH double-labeling was performed as described previously (*Ross, 2011*) with minor modifications. Briefly, testes were embedded in OCT medium, frozen in liquid nitrogen and cryosectioned at 10 µm. Slides were stained with COX buffer for 25 min at RT in the dark. Slides were washed twice with dH20 for 5 min then stained with SDH buffer at 37C for 45 min in the dark. Slides were washed twice with dH20 and destained using a 30%–90–30% acetone gradient. After two additional washes in dH20, slides were counterstained with DAPI and mounted using Fluorogel. For COX/SDH double labeling of sperm, cells were isolated as described above. 20–50 µl of isolated sperm were smeared onto glass slides and allowed to air dry for 1 hr at RT prior to performing the enzymatic assay as described above.

## Fluorescence-activated cell sorting (FACS)

Cells were sorted at Caltech's Flow Cytometry Facility using a Sony SY3200 analyzer. Testes were dissociated from adult males (2–3 months old) as described previously (*Gaysinskaya and Bortvin, 2015*; *Gaysinskaya et al., 2014*). The cell suspension was passed through a 100 µm nylon cell strainer, pelleted at 150 *g* for 5 min, and stained for 1 hr with 5 µg/ml Hoechst 33342 in Flow Cytometry Buffer (HBSS with 2.5 mg/ml fraction V BSA, 10 Mm HEPES buffer, 6.6 mM sodium pyruvate, 0.05% sodium lactate, DNase and 1 mM MgCl2, pH 7.2). The nuclear dye was washed away with Flow Cytometry Buffer, and the cells were filtered through a 40 µm mesh. 7-AAD was added for exclusion of dead cells. Diploid, tetraploid, and haploid cells were isolated as described before (*Bastos et al., 2005*), and verified using germ cell-specific markers *Figure 3—figure supplement 1*.

## Respiration measurements

For respiration measurements in dissociated germ cells, testicular cells were sorted using FACS as described above into germ cell collection media (DMEM 11995 supplemented with 10% FBS, 1% Pen/Strep, 6.6 mM sodium pyruvate, 0.05% sodium lactate). The cells were pelleted at 150 g for 5 min and incubated in Seahorse media (Sigma-Aldrich; #D5030 supplemented with 5% FBS, 1% Pen/Strep, 2 mM glutamine, 6.6 mM sodium pyruvate, 3 mM sodium lactate, and 10 mM glucose) before plating onto 96-well plates pre-coated with Cell-Tak (Corning; 354240) per manufacturer's instructions. The plated cells were spun for 1000 rpm for 5 min to promote cell-adhesion. The Mitochondrial Stress Test was performed using a Seahorse Biosciences Extracellular Flux Analyzer (model XF96). 5 µM oligomycin was added to inhibit complex V; 10 µM CCCP was added to uncouple the proton gradient; and 5 µM Antimycin A was added to inhibit complex III. For respiration measurements in immortalized spermatocytes (GC2spd(ts)), a non-targeting cell line and two separate shMfn1;Mfn2 cell lines were used: shMfn1;Mfn2 (A) and shMfn1;Mfn2 (B). 20,000 cells were plated onto 96-well plates 16 hr before the Seahorse experiment in complete media (DMEM 11995 supplemented with 10% FBS, 1% Pen/Strep). One hour before the experiment, cells were washed into Seahorse media (Sigma-Aldrich; #D5030 supplemented with glutamine, sodium pyruvate, Pen/Strep, and 25 mM glucose). The Mitochondrial Stress Test was performed as described above. Values were normalized to the total number of cells remaining per well after the Seahorse experiment. For counting, cells were trypsinized, resuspended in complete medium, and counted using a tally counter. Normalization was performed using Agilent's Seahorse Wave Desktop Software. shRNA knockdown in immortalized spermatocytes.

GC-2spd(ts) immortalized spermatocytes (CRL-2196) were purchased from ATCC. For dual Mfn1 and Mfn2 knockdown in immortalize spermatocytes, shRNAs against and Mfn1 and Mfn2 were cloned into the FUChW-H1H1 vector containing dual H1 promoters (*Rojansky et al., 2016*) and transferred to pRetroX. The shRNA target sequences are below.

Non Targeting: CGTTAATCGCGTATAATACGC
Mfn1: CGGTATCTCCACTGAAGCA
Mfn2-1: GGATTGGTCATACCACCAATT
Mfn2-2: GCCTGGATGCTGATGTGTTTG

Retroviral vectors were cotransfected into 293 T cells with pCL-Eco (Adgene #12371) using calcium phosphate precipitation. Virus was collected, filtered, and added to GC-2spd(ts) immortalized spermatocytes in the presence of Polybrene. Cells were spun at 2,400 rpm for 30 min and incubated for 8 hr before replacing with complete medium (DMEM 11995, 10% fetal bovine serum, 1% Pen/Strep). To achieve efficient Mfn2 KD, spermatocytes were first dually knocked down for Mfn1 and Mfn2 using shMfn1;Mfn2-1, followed by Mfn2 knockdown using shMfn2-2).

## Stable isotope labeling of amino acids in cell culture (SILAC)

DMEM lacking arginine and lysine was used, along with 10% dialyzed fetal bovine serum. For heavy labeling, Arg6 (U-$^{13}C_6$) and Lys8 (U-$^{13}C_6$, U-$^{15}N_2$) (Cambridge Isotopes) were supplemented at the same concentration as in the standard DMEM formulation. For light labeling, regular DMEM was used.

## Isolation of mitochondria

Mitochondria were isolated from a 1:1 mixture of heavy and light SILAC-labeled, WT and *Mfn1/Mfn2*-null MEFs. Cells were lysed using a nitrogen bomb (Parr) at 200 psi for 10 min. Lysates were centrifuged at 600 *g* for 5 min to remove nuclei and intact cells. The supernatant was resuspended in 15 ml Isolation Buffer (IB) (220 mM Mannitol, 70 mM sucrose, 10 mM HEPES-KOH pH 7.4, 1 mM EGTA, and protease inhibitors) and homogenized with a glass–glass Dounce homogenizer. Lysates were centrifuged for 600 *g* for 5 min and the supernatants containing the crude mitochondrial fraction were centrifuged at 10,000 *g* for 10 min. Crude mitochondria were further purified by a discontinuous Percoll gradient consisting of 80, 52% and 21% Percoll. Following centrifugation at 23,200 rpm for 90 min, mitochondria were collected at the 21%/52% interface, washed with IB, and pelleted by centrifugation at 13,000 rpm for 10 min.

## Mass spectrometry

Samples were analyzed on a hybrid LTQ-Orbitrap mass spectrometer with a nanoelectrospray ion source (Thermo Scientific) coupled to an EASY-nLC HPLC (Proxeon Biosystems, Waltham, MA). Peptides were separated online using a 75 μm ID by 15 cm silica analytical column packed in-house with reversed phase ReproSil-Pur C18AQ 3 μm resin (Dr. Maisch GmbH, Ammerbuch-Entringen, Germany). The analytical gradient was 2–30% solvent B in 150 min at 350 nl/min where solvent B was acetonitrile in 0.2% formic acid. The mass spectrometer collected spectra in data-dependent mode, with a survey spectrum collected from 300 to 1700 m/z at 60 k resolution followed by 10 MS/MS spectra from the most intense precursors. Precursors were fragmented using CID with 35% normalized collision energy and activation Q of 0.25. Only precursors with charge states 2+ and 3+ were selected for fragmentation and dynamic exclusion was enabled with duration of 90 s and an exclusion window of 10 ppm.

## Bioinformatical analysis

Thermo raw files were searched using MaxQuant (v. 1.6.1.0) (*Cox and Mann, 2008*; *Cox et al., 2011*) against the UniProt mouse database (61,681 entries) and a contaminant database (245 entries). Arg6 and Lys8 SILAC labels were specified with re-quantify and match between runs enabled. Trypsin was specified as the digestion enzyme with up to two missed cleavages allowed. Oxidation of methionine and protein N-terminal acetylation were specified as variable modifications and carbamidomethylation of cysteine was specified as a fixed modification. Precursor mass tolerance was 4.5 ppm after recalibration and fragment ion mass tolerance was 0.5 Da.

Following the database search, proteins not annotated in MitoCarta (*Calvo et al., 2016*) were removed. Protein replicate ratios were averaged together and these ratios were then normalized so that the global mean ratio was 1:1. Limma was used to assess individual protein fold change significance and confidence intervals (*Ritchie et al., 2015*; *Smyth, 2004*). One-sample t-tests were used to identify GO terms with annotated protein mean ratios statistically different from 1:1. All *p*-values were adjusted using the Benjamini and Hochberg method (*Benjamini and Hochberg, 1995*).

### Replicates and statistical reporting

Pairwise comparisons were made using the Student's t-test. When multiple pairwise comparisons were made from the same dataset, *p*-values were adjusted using the Bonferroni correction. For comparisons of more than two means, one-way ANOVA was used, followed by Tukey's post hoc test. Number of mice and replicates are indicated in figure legends. All outliers were included in the analysis. All data are represented as mean ± SEM. **** indicates $p \leq 0.0001$; *** indicates $p \leq 0.001$; ** indicates $p \leq 0.01$; * indicates $p \leq 0.05$.

## Acknowledgements

We thank Hsiuchen Chen for her initial characterization of S8::Mfn1 and S8::Mfn2 mice, for help with maintaining mouse colonies, and for overall guidance throughout the project. We thank Safia Malki and Alex Bortvin for providing a detailed protocol for chromosomal spreading and with help identifying germ cell-specific markers; Prabhakara P Reddi for providing the SP-10 antibody; and Jared Rutter for advice on studying MPC1. We thank all members of the Chan Lab for helpful discussions and for comments on the manuscript. Grigor Varuzhanyan was supported by a National Science Foundation Graduate Research Fellowship (DGE-1144469) and a National Institutes of Health Cell and Molecular Biology Training Grant (GM07616T32). Sonja Hess and Robert LJ Graham were supported by grants from the Gordon and Betty Moore Foundation through GBMF775 and the Beckman Institute. This work was supported by NIH grants GM119388 and GM127147. MSL was supported by the National Institute of Allergy and Infectious Diseases (NIAID) (2 P50 AI150464) (to Pamela J Bjorkman, Caltech). We thank the Caltech Kavli Nanoscience Institute for maintenance of the TF-30 electron microscope.

## Additional information

### Funding

| Funder | Grant reference number | Author |
|---|---|---|
| National Institutes of Health | R35 GM127147 | David C Chan |
| National Institutes of Health | GM119388 | David C Chan |
| National Institutes of Health | Cell and Molecular Biology Training Grant (GM07616T32) | Grigor Varuzhanyan |
| National Science Foundation | Graduate Research Fellowship (DGE-1144469) | Grigor Varuzhanyan |
| Gordon and Betty Moore Foundation | GBMF775 | Robert LJ Graham Sonja Hess |
| Beckman Institute | | Robert LJ Graham Sonja Hess |
| National Institute of Allergy and Infectious Diseases | 2 P50 AI150464 | Mark S Ladinsky |

The funders had no role in study design, data collection and interpretation, or the decision to submit the work for publication.

## Author contributions
Grigor Varuzhanyan, Conceptualization, Investigation, Methodology, Writing—original draft, Writing—review and editing; Rebecca Rojansky, Mark S Ladinsky, Investigation; Michael J Sweredoski, Robert LJ Graham, Formal analysis; Sonja Hess, Supervision; David C Chan, Conceptualization, Supervision, Funding acquisition, Writing—review and editing

## Author ORCIDs
Grigor Varuzhanyan (iD) https://orcid.org/0000-0001-6165-0857
Michael J Sweredoski (iD) http://orcid.org/0000-0003-0878-3831
David C Chan (iD) https://orcid.org/0000-0002-0191-2154

## Ethics
Animal experimentation: This study was performed in strict accordance with the recommendations in the Guide for the Care and Use of Laboratory Animals of the National Institutes of Health. All of the animals were handled according to approved institutional animal care and use committee (IACUC) protocols of the California Institute of Technology.

## Decision letter and Author response
Decision letter https://doi.org/10.7554/eLife.51601.sa1
Author response https://doi.org/10.7554/eLife.51601.sa2

## Additional files

### Supplementary files
• Supplementary file 1. Table showing enriched Gene Ontology terms from SILAC experiments in MEFs. SILAC was performed on mitochondria isolated from WT or *Mfn1/Mfn2*-null MEFs. Related to *Figure 6A*.

• Supplementary file 2. Table showing the SILAC ratios for individual mitochondrial proteins. Related to *Figure 6B*.

• Transparent reporting form

### Data availability
All data generated or analyzed during this study are included in the manuscript and supporting files.

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
