## [Decision Letter]

[Editors’ note: a previous version of this study was rejected after peer review, but the authors submitted for reconsideration. The first decision letter after peer review is shown below.]

Thank you for submitting your work entitled "Mitochondrial fusion is required for multiple stages of mouse spermatogenesis" for consideration by *eLife*. Your article has been reviewed by three peer reviewers, one of whom is a member of our Board of Reviewing Editors, and the evaluation has been overseen by a Senior Editor. The following individuals involved in review of your submission have agreed to reveal their identity: Martin Picard (Reviewer #2); Tsutomu Endo (Reviewer #3).

Our decision has been reached after consultation between the reviewers. Based on these discussions and the individual reviews below, we regret to inform you that your work will not be considered further for publication in *eLife*.

The reviewers have made so many suggestions for making the manuscript more complete, that we do not see any way all these comments can be addressed in the time frame of a major revision. Of course, if you wish to make all suggested changes and resubmit, the same reviewers could look through the manuscript again. However, such a resubmission will be treated as new and we could not guarantee that other new questions will not arise during review.

As you can see from the appended reviews, it was the reviewers' opinion that:

1) The presented data nicely documents the spermatogenesis problem related to the absence of mitofusins, but that the data are essentially phenomenological in that they do not show how and why exactly mitofusin deletion causes spermatogenesis defect at the molecular level.

2) Given the fact that mitofusins, especially mitofusin 2, is believed to be implicated in functions other than mitodynamics (e.g. MAM), it is crucial that you document and compare the spermatogenesis phenotype(s) for the single knockout lines that you already have.

3) It is not clear whether the reported problem in spermatogenesis is due to a generic mitodynamic alteration or specifically linked to some intrinsic property of mitofusins. It would thus be critical that you show if deletion of OPA (may be difficult given the multiple isoforms) or overexpression of DRP1 can phenocopy the effects of mitofusin deletion on spermatogenesis.

4) While the idea to run a proteomic experiment to gain insights into the molecular underpinning of the observed defect is exciting, neither the results nor the selection of the cell type (i.e. MEFs) are particularly compelling. Indeed, the reported proteomic changes are quite generic and MEFs do not respect the cell-specific context of spermatocytes.

5) Several comments also focused on the quality of the EM and the data generated with these images in terms of apoptosis and mitochondria cristae, which will need to be corrected.

*Reviewer #1:*

In this new manuscript the authors seek to determine the role of mitofusins in spermatogenesis. This work is prompted by the observation that low Mfn2 expression is associated with reduced sperm motility and Mnf1 defects are associated with alerted spermatogenesis. However, at this time, very little is known about the link of mitofusins with spermatogenesis. Using conditional knockout mouse line deficient in either or both Mfn, the authors found that mutant mice exhibited small testis with reduced spermatozoa in the epididymides with the following severity arcos the different null lines: Mfn2 < Mfn1 < Mfn1/2. In light of this finding the authors then demonstrated in absence of Mfn the tubules of the double knockout mice were almost devoid of post-meiotic spermatids, which they demonstrate to be due to a blockage at the zygotene to pachytene transition. The authors demonstrate that at the pachytene stage, mitochondria are more clustered, their content is higher and their length is greater in spermatocytes, than at the zygotene stage. In keeping with these differences, spermatocytes in the double knockout mice have less mitochondria and they are more fragmented. Moreover, compared to the mitochondria in the wild-type mice spermatocytes, in the knockout mice many mitochondria lack MPC1 which is necessary to import the Sertoli cell-derived lactate used by spermatocytes to fuel OXPHOS. Together with functional OXPHOS measurements, the authors concluded that the above findings indicate that during the zygotene to pachytene transition, mitochondria undergo a metabolic shift, which requires a presence of Mfns. The authors also demonstrated that if the defect of Mfn is chronic, they observed a major depletion in germ cells, which likely die by activation the molecular program of apoptosis. Lastly, the authors utilize SILAC in MEFs isolated mitochondria from the double knockout mice. This proteomic approach revealed that the lack of Mfns is associated with a downregulation of several mitochondrial respiratory chain and mitochondrial ribosomal proteins, as well as TFAM. Conversely several protein import factors were upregulated in these mutant cells. In light of these results the authors conclude that they elucidate how Mfns promote fertility.

Overall this study is well conducted and the results clear and, for the most part convincing. The methods are appropriate although the statistical treatment of the data may require some clarification (see below). The enthusiasm for this work is however reduced, as for this reviewer, as the author's strong conclusion is not supported by the data, and this is discussed below.

1) While the rationale for this work is somewhat tenuous (relying on limited number of observations), the set of investigations presented herein provide a strong case for the hypothesized role of Mfns in spermatogenesis. Yet, the data are essentially phenomenological and do not, in this reviewer's opinion, "…elucidate how Mfns promote fertility" nor provide clear molecular data to explain how Mfns interfere with spermatogenesis. What the authors show is that in absence of Mfns, there is a block in the spermatogenesis if Mnfs are deficient but why that is, unless overlooked, is not demonstrated.

2) It is also unclear whether the deficit in Mfn1 or Mfn2 alone cause the same problem, even if the first set of experiments suggest so.

3) If we focus on Mfns role in mitochondrial dynamics, one may wonder if the effect of Mfns deficit reported here are specific of these proteins or if altering any other mitochondrial dynamic factors would cause the same set of mitochondrial and spermatogenesis problems.

4) If one accepts that Mfn, especially Mfn2, is also involved in other mitochondrial functions than dynamics (e.g. tethering of the mitochondrial-associated membrane), it would be important to at least discuss these other possibilities.

5) The illustration of apoptosis even in EM are to this reviewer not convincing. Would it be possible to select more convincing images or perform additional experiments using functional markers of apoptosis such as activated caspases or cleaved fragments from substrate of activate caspase?

6) Most bar graphs show several groups (and several variables, e.g. time and genotype) suggesting that several t-tests were run per data set. Would it not be then necessary for the others to correct the null-hypothesis rejection threshold by applying, for example, a Bonferroni correction? Even better, in light of the data, would it not be better to use a 1 or 2-way ANOVA followed by a post-hoc test? Indeed, in the case of two factors are include in the investigations (e.g. time and genotype) it might have been ideal to run a 2-way ANOVA to be able to assess the interaction between the two factors.

7) This reviewer is confused by the fact that the authors state in the subsection “Replicates and Statistical Reporting”, "at least four biological replicates" and then "at least three technical replicates" and then "N… always ≥10." Please clarify.

*Reviewer #2:*

The rationale for the study and for spermatogenesis defects in mitochondrial disorders is well-developed and is an important problem that remains unresolved.

For the EM analysis in Figure 6C-E, it may be valuable to quantify the proportion of cristae with and without cristae junctions. Vesicular cristae or compartments have been shown to exist in greater abundance in disease conditions caused by both defects in MICOS complex proteins, and primary respiratory chain defects.

The presentation of cristae junction diameter as frequency distribution is very useful. Is Figure 5C EM tomography, or transmission EM?

Details for the isolation of mitochondria (centrifugation speeds and duration, etc.) should be provided. This could influence which population of mitochondria are isolated and their purity.

Although I have not seen the previous reviews, the author's arguments regarding the cell autonomous/non-autonomous nature of the process are sound.

*Reviewer #3:*

This manuscript describes the importance of mitochondria fusion during spermatogenesis. By using a mouse model, the authors show that germ cell-specific depletion of mitofusins (Mfn1, Mfn2, or both Mfn1 and 2) results in defect in meiotic spermatocytes. Moreover, the authors show that long-term depletion of mitofusins causes depletion of mitotic spermatogonia, suggesting the function of mitochondria fusion in spermatogonia. In general, mitochondria function is important in male germ cells after they complete spermatogenesis, as sperm; sperm middle piece contains mitochondria and play a role in energy production for their progressive motility. The emerged concept that mitochondria function is required for early spermatogenesis, especially in spermatogonia and spermatocytes, is novel and exciting. The work represents significant contribution to the better understanding of spermatogenesis, meiosis, and cell biology. The experiments are well conducted. However, the manuscript writing can be improved to convince readers that the authors' logic, flow, and their conclusions are correct.

1) The authors show that Stra8-Cre/Mfn1 and Mfn2 double mutant (S8::Dm) mice have a reduction of pachytene/diplotene spermatocytes and an abundance of leptotene/zygotene spermatocytes (Figure 2), and conclude that mitofusins are required for zygotene-to-pachytene transition. However, abnormal meiotic spermatocytes are usually arrested at middle pachytene stage: this checkpoint mechanism is known as "mid-pachytene arrest" or "(seminiferous) stage IV arrest." Thus, mitofusins might be required for meiotic progression but not for zygotene-to-pachytene transition. The term "zygotene-to-pachytene transition" should be rephrased into "meiotic progression." Otherwise, the authors should show the data that reduction of pachytene spermatocytes occurs at early pachytene stage, but not at middle/late pachytene stages.

2) The authors use Stra8-Cre reporter mice for the germ cell-specific depletion of Mfn1 and/or Mfn2. However, endogenous Stra8 is expressed in late, but not in early stage of undifferentiated spermatogonia including stem cells (Endo et al., 2015, DOI:10.1073/pnas.1505683112; Teletin et al., 2019). Once late undifferentiated spermatogonia express Stra8, the cells are irreversibly destined for differentiation and cannot contribute to the population of undifferentiated spermatogonia or stem cells. Thus, the authors cannot exclude the possibility that mitofusins have major function in undifferentiated spermatogonia. The authors should revise the manuscript, where function of mitofusins in undifferentiated spermatogonia is shown or discussed. Otherwise, the authors should show the evidence that Stra8-Cre is expressed in stem cells, and mitofusins are depleted in these cells, by co-immunostaining with any one of stem cell markers (ID4 or GFRα1).

3) The authors use flow cytometry to isolate "diploid spermatogonia" and "tetraploid MPI spermatocytes" for the measurement of oxidative phosphorylation (OXPHOS) activity (Figures 3F and Supplementary Figure 3B and Figure 3—figure supplement 2). When germ cells are isolated by DNA content (2C, 4C, or 1C), it is unclear if the mitotic spermatogonia underwent DNA replication (4C) is included in the fraction of "diploid spermatogonia" or "tetraploid MPI spermatocyte."

4) In S8::Dm mice, the number of seminiferous tubules contained differentiating spermatogonia is about 30% of control (Supplementary Figure 2B), and the number of seminiferous tubules contained leptotene/zygotene spermatocytes is about 55% of control (Figure 2E). It seems that the reduced number of differentiating spermatogonia is recovered at leptotene/zygotene stage. Interpretation of the data should be incorporated in the manuscript.

5) Figures 2F-G: It is unclear if the spermatocytes of S8::Dm mice at each stage of MPI (leptotene, zygotene, pachytene, and diplotene) are normal or not. The authors should clarify this point. Chromosome spread images in S8::Dm mice can be shown if needed.

---

## [Author Response]

[Editors’ note: the author responses to the first round of peer review follow.]

[…] As you can see from the appended reviews, it was the reviewers' opinion that:1) The presented data nicely documents the spermatogenesis problem related to the absence of mitofusins, but that the data are essentially phenomenological in that they do not show how and why exactly mitofusin deletion causes spermatogenesis defect at the molecular level.

We agree that understanding molecular mechanism is important. In new data, we have extended our proteomic data to a spermatocyte cell line (Figures 7 and Figure 7—figure supplements 1-3) and germ cells in vivo (Figures 8 and Figure 8—figure supplements 1-3). In brief, we show that mitofusin loss results in reduced levels of OXPHOS subunits and OXPHOS activity. These metabolic defects prevent two key transitions during spermatogenesis: spermatogonial differentiation and meiosis. The new data are described in greater detail in the response to reviewer #1 below.

2) Given the fact that mitofusins, especially mitofusin 2, is believed to be implicated in functions other than mitodynamics (e.g. MAM), it is crucial that you document and compare the spermatogenesis phenotype(s) for the single knockout lines that you already have.

Mitofusin 2 has indeed been implicated in mediating connections between the endoplasmic reticulum and mitochondria. The revised manuscript has detailed analysis of the single mitofusin mutants. The key point of the new data is that loss of Mfn1 alone results in defects in both meiosis and spermatogonial differentiation, thereby mimicking the main features of the double knockout (Figure 2—figure supplement 1 and Figure 4—figure supplements 1-44). These observations argue against an Mfn2-specific function being responsible for the double knockout phenotype.

3) It is not clear whether the reported problem in spermatogenesis is due to a generic mitodynamic alteration or specifically linked to some intrinsic property of mitofusins. It would thus be critical that you show if deletion of OPA (may be difficult given the multiple isoforms) or overexpression of DRP1 can phenocopy the effects of mitofusin deletion on spermatogenesis.

In correspondence with the editors, it was decided that analysis of a completely new mouse knockout (like an *Opa1* knockout) would be beyond the scope of this work, as long as we address the issue of the single mitofusin mouse knockouts. It was also suggested that we acknowledge in the Discussion that the link to mitochondrial fusion is based on the analysis of mitofusins only. We have done this by analyzing the single mitofusins knockouts in detail (as mentioned above) and including the following in the Discussion:

"The two major phenotypes of S8::Dmmice – meiotic failure and spermatogonial differentiation – are also observed with loss of *Mfn1* alone. […] To further test that our spermatogenesis defects are due to loss of mitochondrial fusion, it will be helpful to analysis mice lacking other mitodynamics factors, particularly *Opa1*."

4) While the idea to run a proteomic experiment to gain insights into the molecular underpinning of the observed defect is exciting, neither the results nor the selection of the cell type (i.e. MEFs) are particularly compelling. Indeed, the reported proteomic changes are quite generic and MEFs do not respect the cell-specific context of spermatocytes.

We understand the objection to assuming that data from MEFs will apply to spermatocytes. In the revised manuscript, we have added two new lines of work that extend the proteomic data from MEFs into a spermatocyte cell line (Figure 7 and Figure 7—figure supplements 1-3) and spermatocytes in vivo (Figure 8 and Figure 8—figure supplements 1-3). In both systems, we show reduced and heterogenous expression of OXPHOS subunits. In addition, we go beyond the protein level to show functional consequences, such as reduced OXPHOS activity.

5) Several comments also focused on the quality of the EM and the data generated with these images in terms of apoptosis and mitochondria cristae, which will need to be corrected.

We have thought about the reviewer's concern about the use of EM data to show apoptosis. It is difficult to use EM to quantify apoptosis, unless large numbers of micrographs are used. Because the main issue concerns apoptosis, we decided to firm up our data on apoptosis by performing additional TUNEL staining, which is the most common way to quantify apoptosis. The new data are shown in Figure 5E, F, and the additional analysis has resulted in stronger statistical significance.

We have also reanalyzed the data on mitochondrial cristae, and these changes are discussed in the response to reviewer #2. In particular, we show that the vesiculated cristae structures all connect to the inner membrane.

Reviewer #1:[…] Overall this study is well conducted and the results clear and, for the most part convincing. The methods are appropriate although the statistical treatment of the data may require some clarification (see below). The enthusiasm for this work is however reduced, as for this reviewer, as the author's strong conclusion is not supported by the data, and this is discussed below.1) While the rationale for this work is somewhat tenuous (relying on limited number of observations), the set of investigations presented herein provide a strong case for the hypothesized role of Mfns in spermatogenesis. Yet, the data are essentially phenomenological and do not, in this reviewer's opinion, "…elucidate how Mfns promote fertility" nor provide clear molecular data to explain how Mfns interfere with spermatogenesis. What the authors show is that in absence of Mfns, there is a block in the spermatogenesis if Mnfs are deficient but why that is, unless overlooked, is not demonstrated.

We agree it is important to identify the molecular mechanism underlying the block in spermatogenesis. The revised manuscript addresses this issue by extending insights from the proteomics in MEFs into a spermatocyte cell line and spermatocytes in vivo. We found the GC-2spd(ts) spermatocyte cell line to be highly sensitive to mitofusin knockdown (more so than MEFs). Knockdown of Mfn1 and Mfn2 results in mitochondrial fragmentation (Figure 7A, B) and reduction of Complex I subunit NdufB8, Complex Isubunit SDHB, Complex II subunit UQRC2, Complex IV subunit MTCOI, and Complex V subunit ATP5A (Figure 7C, D). In addition to these reductions in protein levels, we find a reduction in OXPHOS activity as measured by Seahorse experiments (Figure 7E-G). Similar knockdown in MEFs was not sufficient to cause OXPHOS changes in MEFs (Figure 7—figure supplement 3D), suggesting that the spermatocyte cell line is particularly vulnerable to loss of mitochondrial fusion.

We found similar defects in mutant spermatocytes in vivo (Figure 8). Spermatocytes have reduced and heterogeneous expression of Complex I subunit NdufB6, Complex IV subunit MTCOI, and the mitochondrial pyruvate carrier MPC1 (Figure 8A, B). We also obtained evidence of reduced OXPHOS activity through histochemical analysis of COXIV/SDH activity (Figure 8C, Figure 8 Figure supplement 3A-B).

These results show that loss of mitofusins results in reduced levels of OXPHOS subunits and reduced OXPHOS activity in spermatocytes. Combined with the observation that meiosis in spermatocytes is normally accompanied by a metabolic shift characterized by increased mitochondrial content and OXPHOS activity, these findings provide a compelling mechanistic explanation for the spermatogenesis defect.

2) It is also unclear whether the deficit in Mfn1 or Mfn2 alone cause the same problem, even if the first set of experiments suggest so.

In the revised manuscript, we provide much more detailed analyses of the Mfn1 and Mfn2 single mutants. We find that loss of Mfn1 alone results in the two major phenotypes of the double mutant: a defect in meiosis and spermatogonial differentiation (Figure 2—figure supplement 2A-B, Figure 4—figure supplement 4A-B). Loss of Mfn2 alone results only in the latter phenotype.

3) If we focus on Mfns role in mitochondrial dynamics, one may wonder if the effect of Mfns deficit reported here are specific of these proteins or if altering any other mitochondrial dynamic factors would cause the same set of mitochondrial and spermatogenesis problems.

In a discussion with the editors, it was agreed that analysis of additional mouse knockouts (such as an Opa1 knockout) is beyond the scope of this paper. However, we acknowledge in the Discussion the importance of studying other proteins involved in mitochondrial fusion to further resolve this issue:

"The two major phenotypes of S8::Dm – meiotic failure and spermatogonial differentiation – are also observed with loss of *Mfn1* alone. This observation suggests that these defects should be attributed to loss of mitochondrial fusion, rather than non-fusion functions, like ER-mitochondria tethering, that have been attributed to Mfn2 (De Brito and Scorrano, 2008). To further test that our spermatogenesis defects are due to loss of mitochondrial fusion, it will be helpful to analysis mice lacking other mitodynamics factors, particularly Opa1."

4) If one accepts that Mfn, especially Mfn2, is also involved in other mitochondrial functions than dynamics (e.g. tethering of the mitochondrial-associated membrane), it would be important to at least discuss these other possibilities.

As noted above, the fact that Mfn1 single knockout reproduces the two key features of the double mitofusin knockout rules out the possibility that the phenotypes are due to an Mfn2- specific function, like ER-mitochondria tethering.

5) The illustration of apoptosis even in EM are to this reviewer not convincing. Would it be possible to select more convincing images or perform additional experiments using functional markers of apoptosis such as activated caspases or cleaved fragments from substrate of activate caspase?

Because the key point is to provide more definitive evidence for apoptosis, we decided to obtain better TUNEL data. We performed additional TUNEL staining of testis sections, and the new data in Figure 5E, F shows clearly that apoptosis is increased in the mutant.

6) Most bar graphs show several groups (and several variables, e.g. time and genotype) suggesting that several t-tests were run per data set. Would it not be then necessary for the others to correct the null-hypothesis rejection threshold by applying, for example, a Bonferroni correction? Even better, in light of the data, would it not be better to use a 1 or 2-way ANOVA followed by a post-hoc test? Indeed, in the case of two factors are include in the investigations (e.g. time and genotype) it might have been ideal to run a 2-way ANOVA to be able to assess the interaction between the two factors.

Thank you for the suggestion to perform these important statistical tests. We have added some of these tests and included a detailed description of the statistical tests performed in the Materials and methods section. Briefly, when multiple pairwise comparisons were made from the same dataset, *p*-values were adjusted using the Bonferroni correction. For comparisons of more than two means, one-way ANOVA was used, followed by Tukey’s post hoc test. Two way ANOVAs were not used, as we did not make any comparisons between more than one independent variable.

7) This reviewer is confused by the fact that the authors state in the subsection “Replicates and Statistical Reporting”, "at least four biological replicates" and then "at least three technical replicates" and then "N… always ≥10." Please clarify.

We agree that the original description was confusing; it was attempting to explain statistical values for all the animal studies in aggregate. In the revised text, we explain the statistics more clearly by providing N values for each separate experiment in the figure legends.

Reviewer #2:The rationale for the study and for spermatogenesis defects in mitochondrial disorders is well-developed and is an important problem that remains unresolved.For the EM analysis in Figure 6C-E, it may be valuable to quantify the proportion of cristae with and without cristae junctions. Vesicular cristae or compartments have been shown to exist in greater abundance in disease conditions caused by both defects in MICOS complex proteins, and primary respiratory chain defects.

This is a good suggestion. We carefully examined all cristae that could be traced back to the inner mitochondrial membrane using 400 nm tomograms and found that all cristae, including the vesiculated ones in S8::Dm mice, contain cristae junctions.

The presentation of cristae junction diameter as frequency distribution is very useful. Is Figure 5C EM tomography, or transmission EM?

The original Figure 5C was EM tomography. However, we have removed these images due to the issue raised by reviewer #1.

Details for the isolation of mitochondria (centrifugation speeds and duration, etc.) should be provided. This could influence which population of mitochondria are isolated and their purity.

In the revised manuscript, we now provide detailed information about the isolation of mitochondria.

Although I have not seen the previous reviews, the author's arguments regarding the cell autonomous/non-autonomous nature of the process are sound.Reviewer #3:[…] The emerged concept that mitochondria function is required for early spermatogenesis, especially in spermatogonia and spermatocytes, is novel and exciting. The work represents significant contribution to the better understanding of spermatogenesis, meiosis, and cell biology. The experiments are well conducted. However, the manuscript writing can be improved to convince readers that the authors' logic, flow, and their conclusions are correct.1) The authors show that Stra8-Cre/Mfn1 and Mfn2 double mutant (S8::Dm) mice have a reduction of pachytene/diplotene spermatocytes and an abundance of leptotene/zygotene spermatocytes (Figure 2), and conclude that mitofusins are required for zygotene-to-pachytene transition. However, abnormal meiotic spermatocytes are usually arrested at middle pachytene stage: this checkpoint mechanism is known as "mid-pachytene arrest" or "(seminiferous) stage IV arrest." Thus, mitofusins might be required for meiotic progression but not for zygotene-to-pachytene transition. The term "zygotene-to-pachytene transition" should be rephrased into "meiotic progression." Otherwise, the authors should show the data that reduction of pachytene spermatocytes occurs at early pachytene stage, but not at middle/late pachytene stages.

We agree that we do not want to overstate the preciseness of the meiotic block. We have followed the reviewer's suggestion and replaced the term "zygotene-to-pachytene transition" with "meiotic progression".

2) The authors use Stra8-Cre reporter mice for the germ cell-specific depletion of Mfn1 and/or Mfn2. However, endogenous Stra8 is expressed in late, but not in early stage of undifferentiated spermatogonia including stem cells (Endo et al., 2015, DOI:10.1073/pnas.1505683112; Teletin et al., 2019). Once late undifferentiated spermatogonia express Stra8, the cells are irreversibly destined for differentiation and cannot contribute to the population of undifferentiated spermatogonia or stem cells. Thus, the authors cannot exclude the possibility that mitofusins have major function in undifferentiated spermatogonia. The authors should revise the manuscript, where function of mitofusins in undifferentiated spermatogonia is shown or discussed. Otherwise, the authors should show the evidence that Stra8-Cre is expressed in stem cells, and mitofusins are depleted in these cells, by co-immunostaining with any one of stem cell markers (ID4 or GFRα1).

This is a very interesting point about the apparent discrepancy between endogenous *Stra8* expression and the Cre expression in this transgene. We have followed the reviewer's advice and used GFRα1 as a reporter to monitor expression of the *Stra8-Cre* driver. The results were unambiguous: *Stra8-Cre* is

expressed in nearly 100% of GFRα1-positive spermatogonia (Figure 1—figure supplement 1 and Figure 1—figure supplement 2A S1B and S1C). Furthermore, we show that despite *Stra8-Cre* expression in these early undifferentiated spermatogonia, the mutant cells show Ki67 staining. In addition, the number of GFRα1- positive spermatogonia are increased in the mutant (Figures 4F, G). Thus, these data indicate that mitofusins are dispensable for proliferation of early, undifferentiated spermatogonia.

3) The authors use flow cytometry to isolate "diploid spermatogonia" and "tetraploid MPI spermatocytes" for the measurement of oxidative phosphorylation (OXPHOS) activity (Figures 3F and Supplementary Figure 3B and Figure 3—figure supplement 2). When germ cells are isolated by DNA content (2C, 4C, or 1C), it is unclear if the mitotic spermatogonia underwent DNA replication (4C) is included in the fraction of "diploid spermatogonia" or "tetraploid MPI spermatocyte."

This is an important point. To address this issue, we used molecular markers to characterize the identity of the 4N population after sorting. Figure 3—figure supplement 1 shows that the vast majority (>95%) of 4N cells express the MPI-specific SCP3 marker. Thus, the contribution of replicating spermatogonia to the 4N population is insignificant.

4) In S8::Dm mice, the number of seminiferous tubules contained differentiating spermatogonia is about 30% of control (Supplementary Figure 2B), and the number of seminiferous tubules contained leptotene/zygotene spermatocytes is about 55% of control (Figure 2E). It seems that the reduced number of differentiating spermatogonia is recovered at leptotene/zygotene stage. Interpretation of the data should be incorporated in the manuscript.

This is an interesting observation. In the revised manuscript, we have re-formatted the data presentation so that the two sets of data referred to are in the same plot (Figure 2E) and can be readily compared. The data do suggest an upward trend going from differentiating spermatogonia to leptotene/zygotene. However, the difference was not statistically significant, and we point this out in the figure legend.

5) Figures 2F-G: It is unclear if the spermatocytes of S8::Dm mice at each stage of MPI (leptotene, zygotene, pachytene, and diplotene) are normal or not. The authors should clarify this point. Chromosome spread images in S8::Dm mice can be shown if needed.

We have revisited this issue and find no obvious chromosome abnormalities in S8::Dmspermatocytes. This point has been clarified in the updated manuscript. It should be noted that although we could find no chromosomal abnormalities in S8::Dmspermatocytes, mutant spermatocytes display reduced mitochondrial content (Figures 3A and 3B), aberrant mitochondrial morphology (Figures 3C-3D, 5A-D), downregulation of OXPHOS components (Figures 8A-8B) and reduced OXPHOS activity (Figure 8C).